# Momentum space toroidal moment in a photonic metamaterial

Biao Yang [1,2,10], Yangang Bi[3,4,10], Rui-Xing Zhang [5], Ruo-Yang Zhang [2], Oubo You[3], Zhihong Zhu[1], Jing Feng[4], Hongbo Sun [4,6], C. T. Chan [2✉], Chao-Xing Liu [7✉] & Shuang Zhang [3,8,9✉]

Berry curvature, the counterpart of the magnetic field in the momentum space, plays a vital role in the transport of electrons in condensed matter physics. It also lays the foundation for the emerging field of topological physics. In the three-dimensional systems, much attention has been paid to Weyl points, which serve as sources and drains of Berry curvature. Here, we demonstrate a toroidal moment of Berry curvature with flux approaching to $\pi$ in judiciously engineered metamaterials. The Berry curvature exhibits a vortex-like configuration without any source and drain in the momentum space. Experimentally, the presence of Berry curvature toroid is confirmed by the observation of conical-frustum shaped domain-wall states at the interfaces formed by two metamaterials with opposite toroidal moments.

[1] College of Advanced Interdisciplinary Studies & Hunan Provincial Key Laboratory of Novel Nano-Optoelectronic Information Materials and Devices, National University of Defense Technology, Changsha, China. [2] Department of Physics, The Hong Kong University of Science and Technology, Hong Kong, China. [3] School of Physics and Astronomy, University of Birmingham, Birmingham, UK. [4] State Key Lab of Integrated Optoelectronics, College of Electronic Science and Engineering, Jilin University, Changchun, China. [5] Condensed Matter Theory Center and Joint Quantum Institute, Department of Physics, University of Maryland, College Park, MD, USA. [6] State Key Laboratory of Precision Measurement Technology and Instruments, Department of Precision Instrument, Tsinghua University, Beijing, China. [7] Department of Physics, The Pennsylvania State University, University Park, PA, USA. [8] Department of Physics, The University of Hong Kong, Hong Kong, China. [9] Department of Electrical & Electronic Engineering, The University of Hong Kong, Hong Kong, China. [10] These authors contributed equally: Biao Yang, Yangang Bi. ✉email: phchan@ust.hk; cxl56@psu.edu; s.zhang@bham.ac.uk

**B**erry curvature, a gauge-invariant local manifestation of the geometric properties of the wave functions in the parameter space, has been considered as an essential ingredient in understanding various branches of physics[1–3]. Especially, it has also blossomed into an important research field in periodic crystals as an intrinsic property of their band structures[4,5]. Its role as the "magnetic field" in the momentum space has induced a plethora of significant physical features in the dynamics of Bloch electrons, such as various effects on transports[6], thermodynamics[7,8], and density of states[9] of crystals. In particular, Berry curvature can provide an extra contribution to the group velocity – anomalous velocity[2], for a wave-packet moving in a periodic system. Berry curvature with quantized flux underlies the emerging field of topological physics. Similar to quantized integral of Gaussian curvature over a closed surface, the integration of Berry curvature over a closed surface or a two-dimensional (2D) Brillouin zone in the momentum space is also quantized, giving the unique topological characteristic of a system, the so-called Chern number[10]. Recently, much attention has been paid to the Berry curvature generated by Weyl points[11–16]. A Weyl semimetal contains a minimum of two Weyl points with opposite charges, each serves as the quantized topological monopole that emits or collects the Berry curvature in the three-dimensional (3D) momentum space[11].

The concept of Berry curvature has also been extended from electronic systems to classical fields such as photonics[17], acoustics, and mechanics[18]. In photonics, Berry curvature leads to polarization-dependent light transports in photonic crystals[19] and classical geometrical optics[20]. Very recently, one-way chiral zero modes in Weyl systems under an effective gauge field have also been demonstrated in artificial photonic and phononic metacrystals[21,22].

Meanwhile, toroidal multipolar moments have attracted much attention both in solid state physics and electrodynamics[23,24], with interesting observables including the pronounced toroidal resonances in artificial metamolecules and dielectric nanostructures[25–28]. The excitation of the magnetic toroidal moment is manifested as the configuration of a ring of static or dynamic magnetic field in the real space. They not only exhibit peculiar features in theory but also show promising applications, such as data storage, unique magnetic responses, and interaction with electromagnetic waves[23,24].

Here, we demonstrate the momentum-space toroidal moment (MTM)[23,24] in a photonic metamaterial, where Berry curvature shows a 3D vortex distribution with Berry flux approaching to $\pi$. We further observe helical domain-wall states at the interfaces between two metamaterials with opposite MTMs, which show either positive or negative dispersion depending on the orientations of the metamaterials. The MTM may also lead to observation of various interesting phenomena, such as negative refraction, surface-dependent anomalous shifts, and bulk transverse spin.

## Results

**Gapped topological nodal ring**. To understand the formation of the Berry curvature toroid, we start from the nodal ring phase[29] as shown in Fig. 1a. In a simplified model, the effective Hamiltonian of the nodal ring takes the form, $H_{\mathrm{N}} = (k_x^2 + k_y^2 - m)\sigma_z + k_z\sigma_x$, where $\sqrt{m}$ ($m > 0$) indicates the radius of the nodal ring and $\sigma_i$ is the Pauli matrix. The degeneracy of the two bands occurs on a ring (red) in the $k_z = 0$ plane with a full rotation symmetry around $z$ axis. Considering a 2D plane that contains the $k_z$ axis (here we choose the $k_y - k_z$ plane for illustration without loss of generality), around the gapless point in this 2D plane, the low energy physics can be effectively described by a 2D Dirac Hamiltonian. Consequently, a quantized $\pi$ Berry phase is accumulated along the loop enclosing the gapless point (blue loop in Fig. 1a), which reflects the topological nature of the nodal ring.

Nodal ring usually requires the protection from extra spatial symmetries, such as mirror and inversion. In the presence of symmetry-breaking perturbations, nodal lines may break into several discrete nodal points or become fully gapped[30]. For example, in TaAs[31] nodal lines are gapped into Weyl nodes when spin–orbital coupling is considered. Figure 1b schematically shows the Berry curvature generated by the Weyl points. A different formation of Berry curvature[32] in 3D momentum space arises when the nodal line is fully gapped as shown in Fig. 1c, wherein a rotationally invariant mass term (denoted by the constant $\gamma$ below, i.e., $\propto \gamma\sigma_y$) is introduced to break the mirror symmetry $M_z$ represented by $\sigma_z$. This leads to the emergence of Berry curvature whose distribution in the momentum space forms a toroid due to the rotation symmetry around the $z$ axis, as shown in Fig. 1c. By carrying on the analogue of Berry curvature

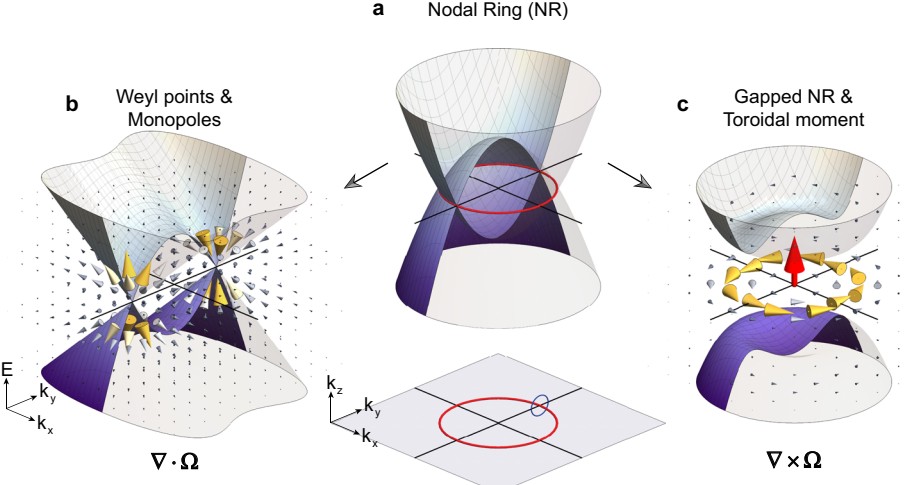

**a** Nodal Ring (NR)

**b** Weyl points & Monopoles

**c** Gapped NR & Toroidal moment

$\nabla \cdot \Omega$ $\nabla \times \Omega$

**Fig. 1 Monopoles and toroidal moments in the momentum space. a** Nodal ring (red ring) and quantized Zak phase (blue loop) protected by mirror symmetry $M_z$. The inset shows the nodal ring in the $k$-space. **b** In Weyl semimetals, paired Weyl points serve as sinks and sources (monopoles) of Berry curvature in the momentum space. **c** Fully gapped nodal ring exhibiting Berry curvature vortex, the momentum-space toroidal moment (MTM).

as the effective magnetic field in the momentum space, this form of the Berry curvature represents the analogue of the toroidal moment in electrodynamics, i.e., a polar toroidal dipole moment[23] $\mathbf{T} \propto \int \mathbf{k} \times \mathbf{\Omega}(\mathbf{k})d^3\mathbf{k}$. Different from the real space static toroidal moments, they require for breaking both time-reversal symmetry $T$ and inversion symmetry $P$. Here $T$ (represented by $\sigma_z K$ with $K$ being complex conjugate) is preserved while $P$ is explicitly broken. In the momentum space both $T$ and $P$ reverse the momentum $\mathbf{k}$, which is different from the real space where only $P$ flips $\mathbf{r}$. Thus, we cannot simply transfer the symmetry classification of electric/magnetic/toroidal dipole moments from real to momentum space. However, the axial or polar nature of various dipole moments does not depend on the space, and both the magnetic toroidal moment in the real space and the Berry curvature toroidal moment in the momentum space are polar vectors.

It is worth noting that for each vertical cutting plane (planes that contain $k_z$ axis), e.g., the $k_y - k_z$ plane, the topological features can be well captured by valley Chern number. Note that the quantization of valley Chern number is exact only when the gap approaches to zero. In fact, the absence of symmetry requirement makes valley-related topological effects more accessible in experimental implementation than those topological phases strictly protected by symmetries for all kinds of waves. So far, 2D valley physics has promised a plethora of applications. Especially in photonics/phononics, researchers have proposed topological laser[33], communications[34], waveguides[35], etc. The valley effects are very robust as long as the gap is small enough[2,6].

**Photonic metamaterials realization**. The MTM can be realized in practice using photonic metamaterials. We start with a previously demonstrated metamaterial that carries nodal line[36] as shown in Fig. 2a, which possesses a single nodal ring in the momentum space that is protected by mirror symmetry $M_z$. The two crossing bands possess different mirror quantum numbers such that they cannot be hybridized. Figure 2c gives the

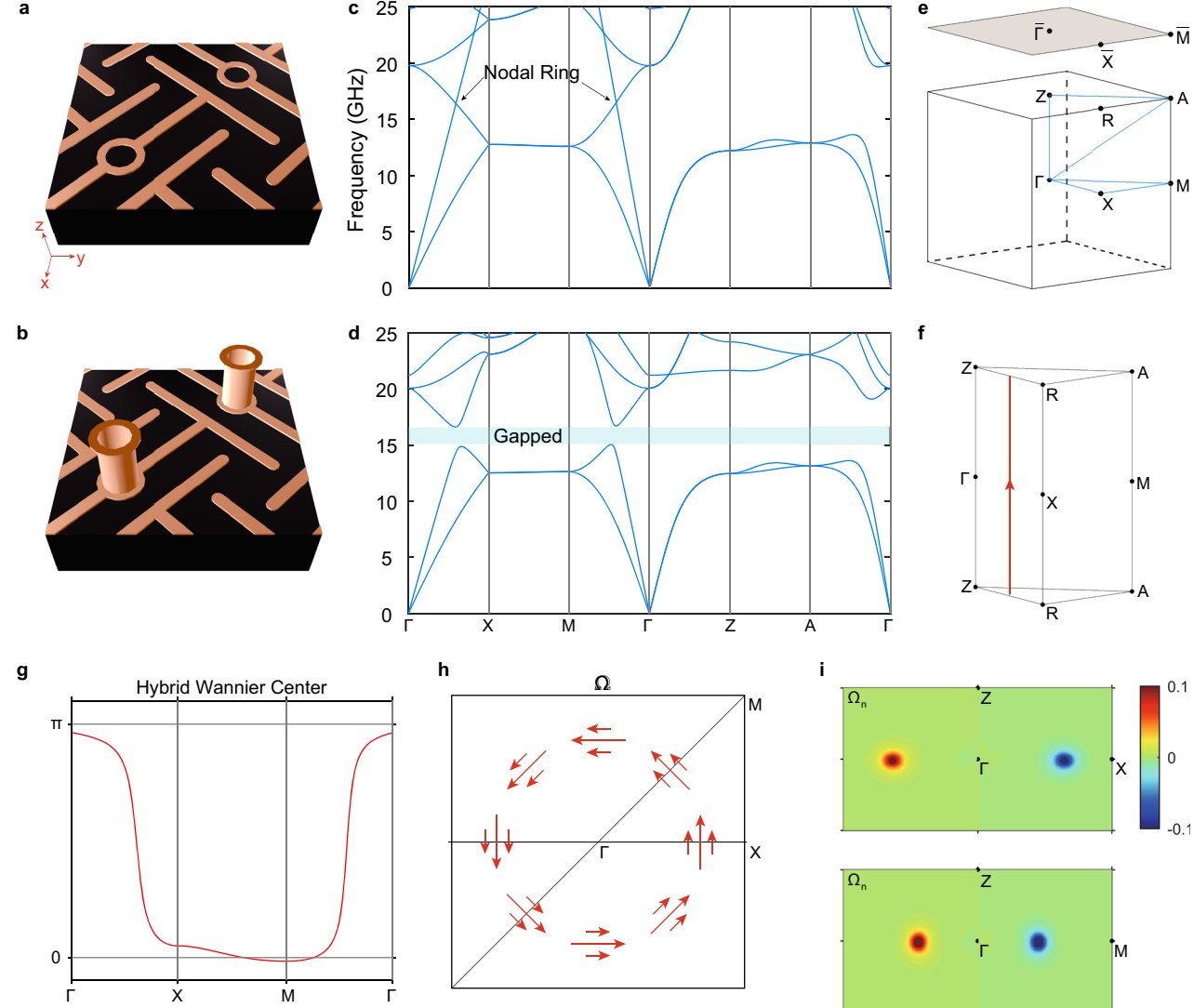

**Fig. 2 Realizing momentum-space toroidal moment (MTM) in a photonic meta-crystal. a** Nodal ring in photonic meta-crystal protected by mirror symmetry $M_z$. **b** Breaking $M_z$ results in a gapped nodal ring. **c**, **d** Band structure along high symmetry lines as defined in **e** for gapless nodal ring and fully gapped nodal ring, respectively. The complete band gap is shadowed. **e** First Brillouin zone (FBZ) with high symmetry points labeled. The momentum path of band structure is indicated explicitly. **f** Reduced FBZ for Wannier center and Berry curvature calculation. **g** Hybrid Wannier center calculated along $\Gamma XM\Gamma$. The Berry phase accumulating direction is indicated by the red line and arrow in **f**. **h** Berry curvature on the $k_z = 0$ plane, which shows the vortex distribution. **i** Calculated Berry curvature distribution on the $k_y = 0$ plane (upper panel) and $k_x = k_y$ plane (lower panel).

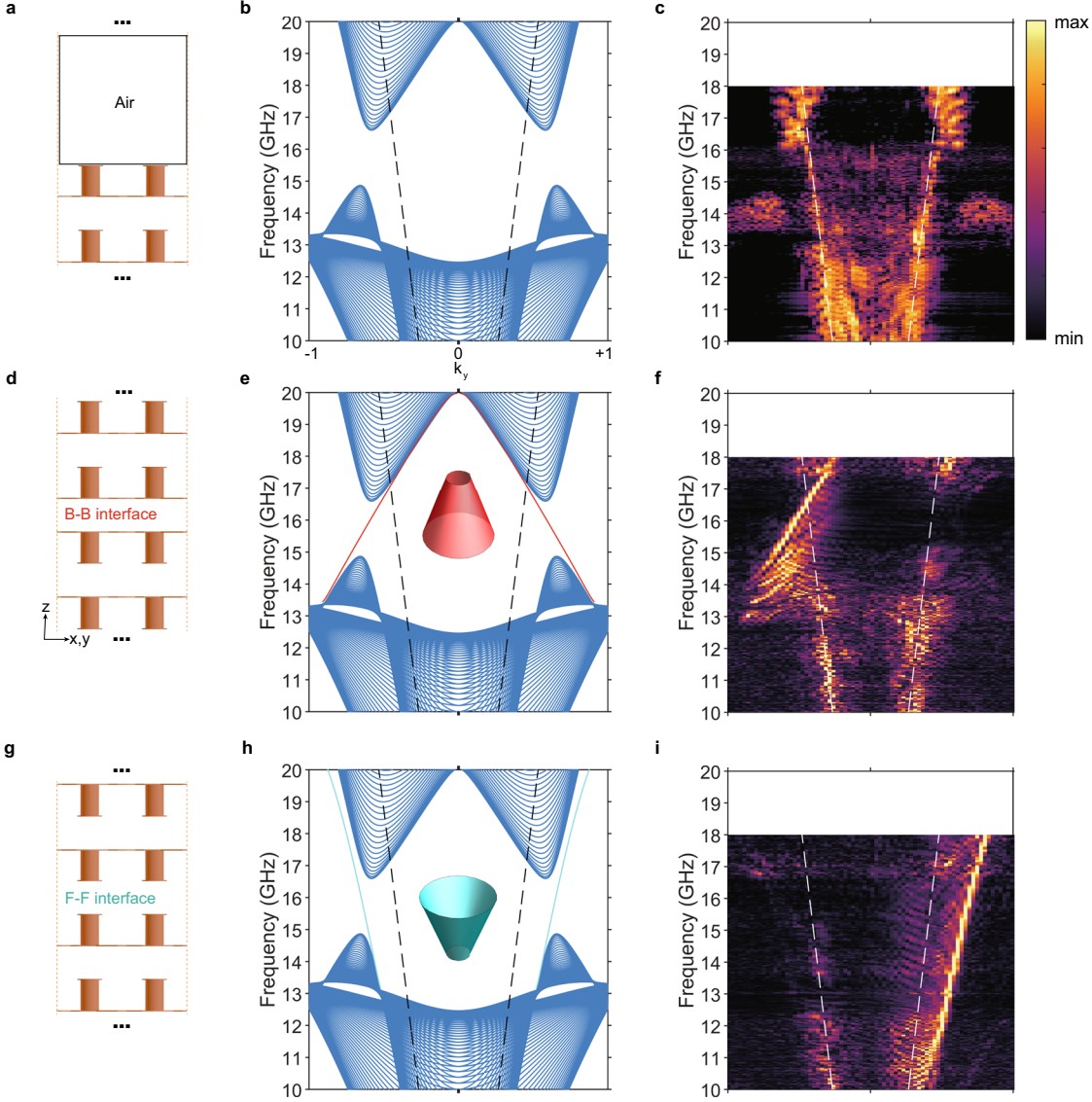

**Fig. 3 Experimentally characterizing interface states between momentum-space toroidal moments (MTMs). a** Configuration for bulk state mapping. There are 20 unit cells used in the experiment. Near field raster-scanning is used to probe the bulk state exponentially decaying tails. **b** Projected bulk bands on the $k_x = 0$ plane with discretized $k_z$. **c** Experimentally mapped bulk states corresponding to the configuration as shown in **a**, where modes in air exist across the complete gap. **d** Configuration for Back–Back (B–B) interface states. On each side there are 10-unit-cell used in the experiment. **e** Back–Back interface states (red solid line) run through the complete gap. The inset shows the 3D view of the interface state. **f** Experimentally mapped Back–Back interface states from Fourier-transforming phase-distribution (See Supplementary Fig. 4). **g–i** Similar to **d–f** but for the Face–Face (F–F) interface. Dashed black/white lines indicate light cones.

corresponding band structure along high symmetry lines in the momentum space with the first Brillouin zone (FBZ) shown in Fig. 2e. By introducing a small metallic bar in the vertical direction to slightly break the mirror symmetry $M_z$ as shown in Fig. 2b, the nodal ring becomes fully gapped, with the resulting band structure presented in Fig. 2d. The underlying topological features of the gapped nodal ring are studied via Wilson loop calculations, which give the Zak phase accumulated along $+k_z$ direction (Fig. 2f, for details see Supplementary Note 1 and Supplementary Fig. 1a). The corresponding hybrid Wannier center for the two bands with frequencies below the bulk bandgap along ΓXMΓ is shown in Fig. 2g. Similar to staggered graphene, the Wannier center approximately approaches to 0 or $\pi$ in the absence of mirror symmetry $M_z$. Applying local Wilson loops with the direction defined in Supplementary Fig. 1b (see Supplementary Note 1), we also obtain the local Berry curvature

distributions on the $k_y = 0$ (upper panel) and $k_x = k_y$ (lower panel) planes for the two bands with frequencies below the bulk bandgap, as shown in Fig. 2i. The Berry curvature is concentrated near the small gap, which corresponds to the sharp slopes of the hybrid Wannier center in Fig. 2g. Figure 2h gives the vortex feature of Berry curvature in the $k_z = 0$ plane. When the gap approaches to zero, it is expected that the Berry curvature becomes more concentrated with flux approaching to $\pm\pi$. On the other hand, with stronger mirror-symmetry breaking, the gap width increases and the Berry curvature distribution becomes more extended. Further increasing of the bandgap results in a less Berry flux integral and a significantly weakened MTM-**T**.

Here the MTM arises from gapping of a nodal ring. Usually for sufficiently small band-gap, the Berry curvature is tightly concentrated around a ring to form the MTM. As the nodal ring does not require the protection of rotation invariance around

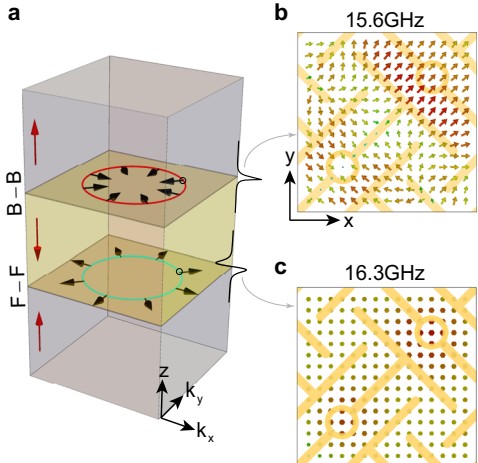

**Fig. 4 Illustration of negative and positive refraction indices of domain wall states at the interfaces. a** Two interfaces (Back–Back and Face–Face) formed by three metamaterials with different orientations indicated by red arrows. The Equi-Frequency Contour (EFC) exhibits negative/positive refraction index of the Back–Back(B–B)/Face–Face(F–F) domain wall state, with group velocity opposite/along the direction of the wave vector. The black arrows indicate group velocity directions. **b, c** Electric field distribution of the Back–Back/Face–Face interface state right at the interface. There are only in-plane components in the longitudinal interface mode (**b**) and out-plane components in the transverse interface mode (**c**), respectively. The corresponding wavevectors are indicated by hollow dots in **a**.

the $z$-axis, the MTM resulting from breaking the mirror symmetry of the nodal line system is not restricted by this constraint either. Thus the $C_4$ rotation symmetry around $z$-axis in our work is not a necessary condition.

**Helical domain-wall states**. Due to the concentration of the Berry curvature at small gaps, it is expected that there exist helical domain wall states[37] with opposite dispersions on the Back-Back (B-B) and Face-Face (F-F) interfaces (cf. Fig. 3d, e, g, h) between two metamaterials with opposite MTMs. In the effective medium limit, due to the rotational symmetry of the system, each Equi-Frequency Contour (EFC) of the domain wall states is a 2D circle. On an arbitrary cutting plane containing the rotation axis, such as the $k_y - k_z$ plane, the reduced 2D system can be regarded as a valley Hall system[6,38], with the integration of Berry curvature over half of the 2D Brillouin zone (e.g., $k_y > 0$) approaching to $\pi$ at very small gaps. The interface states run through the gap and show gapless features, serving as an evidence of the toroidal configuration of the Berry curvature distribution.

Experimentally, the bulk and surface states are investigated with the configurations shown in Fig. 3a, d, g, respectively. The bulk states are measured via raster-scanning the near-field on the interface between the face side (with the metallic protrusions pointing up towards the air) and air as schematically shown in Fig. 3a. Figure 3b shows the projected bands with discretized $k_z$ due to the finite thickness of the sample. The experimental result (Fig. 3c, see details in Methods and Supplementary Fig. 2b) shows strong resemblance to the numerical result shown in Fig. 3b, except for a slight frequency shift. The slight frequency shift may arise from the following reasons: the PCB sample has fabrication errors, while the resonance frequency is very sensitive to the size of metallic structures, such as the height of bars and length of metallic strips; the actual dielectric constant of the PCB substrate materials is slightly dispersive while in the simulation we consider it to be a constant.

A complete gap and the corresponding valleys are clearly visible. Figure 3d, g present the schematic views of Back–Back and Face–Face interface configurations for the investigation of the domain-wall states, respectively. The corresponding simulated interface states are shown in Fig. 3e, h. On the opposite interfaces, these interface states show opposite dispersions. By using the interface line scanning method (see details in Methods and Supplementary Fig. 2c), we experimentally map out the interface states, with the results shown in Fig. 3f, i, respectively. It should be noted that here we used the phase Fourier-transformation results (See Supplementary Fig. 4). The simulation and experimental results fit each other very well (see details in Supplementary Note 2), further confirming the existence of helical interface states, which serves as a direct evidence of the presence of MTM in the designed metamaterial.

In addition to the line-scanning method, we also construct the air-waveguides as shown in Supplementary Figs. 5 and 6, where we insert the probe antenna directly into the 6 mm-thick air-gaps of Face-Face and Back-Back configurations to raster-scan the interface states, respectively. The simulation and experiment results fit well again and further confirm the presence of MTM.

We find when setting the front and back surfaces to be PEC (perfect electric conductor) and PMC (perfect magnetic conductor) boundary conditions, respectively, the front and back surfaces support helical surface states as well (Supplementary Figs. 8 and 9). Wherein, the PEC/PMC boundary condition serving as mirror ($M = -1/+1$) provides the other half Chern number[39] (see details in Supplementary Note 3: surface states with PEC/PMC boundary conditions).

**Negative refraction**. Due to the negative refraction of the interface state on the Back–Back domain wall, the probed field is located mostly on the $-k_y$ region as shown in Fig. 3f. Because the source is located on $y = 0$ (Supplementary Fig. 2c), the energy has to propagate along $+y$ direction. Thus the $-k_y$ region will be excited as they have positive group velocity $v_g$. In electromagnetism the case with ($v_g \cdot k < 0$) corresponds to negative refraction index[40], which has ignited the field of metamaterials with many unconventional applications, such as perfect lens[41]. Here, the two domain-walls separately support negative and positive refraction indices for the interface states as shown in Fig. 4a. Moreover, due to the arising mirror symmetry $M_z$ on the interface, each domain wall state can be labeled with a $M_z$ eigenvalue. Thus they are further classified into longitudinal mode ($M_z = 1$) and transverse mode ($M_z = -1$) as shown in Fig. 4b, c, respectively. With the nearly in-plane isotropic electromagnetic response, the observed domain wall states provide a topological platform[42] for realizing 2D negative/positive refraction index and for exploring the associated applications. To clearly demonstrate the 2D negative refraction, we experimentally construct a Veselago lens configuration[41] and show the ability of focusing light from a point source to a focal point in Supplementary Fig. 7.

**Anomalous shift**. Finally, we calculate the reflection phase on both the front and back surfaces based on effective media theory (see parameters in Supplementary Note 4). The configurations are shown in Fig. 5. We also present the reflection phase for the nodal line system as a reference. The reflection amplitudes are all unity as the selected frequency is located in the gap or right at the nodal degeneracy (Fig. 5a, b), whereas only the phases of reflection change with the in-plane wavevector $k_x$. A TM (transverse magnetic with $H_z = 0$, indicated as red lines in Fig. 5c–e) beam is incident onto the front/back surface. Figure 5f–h shows the dependence of reflection phases on the in-plane wavevector for

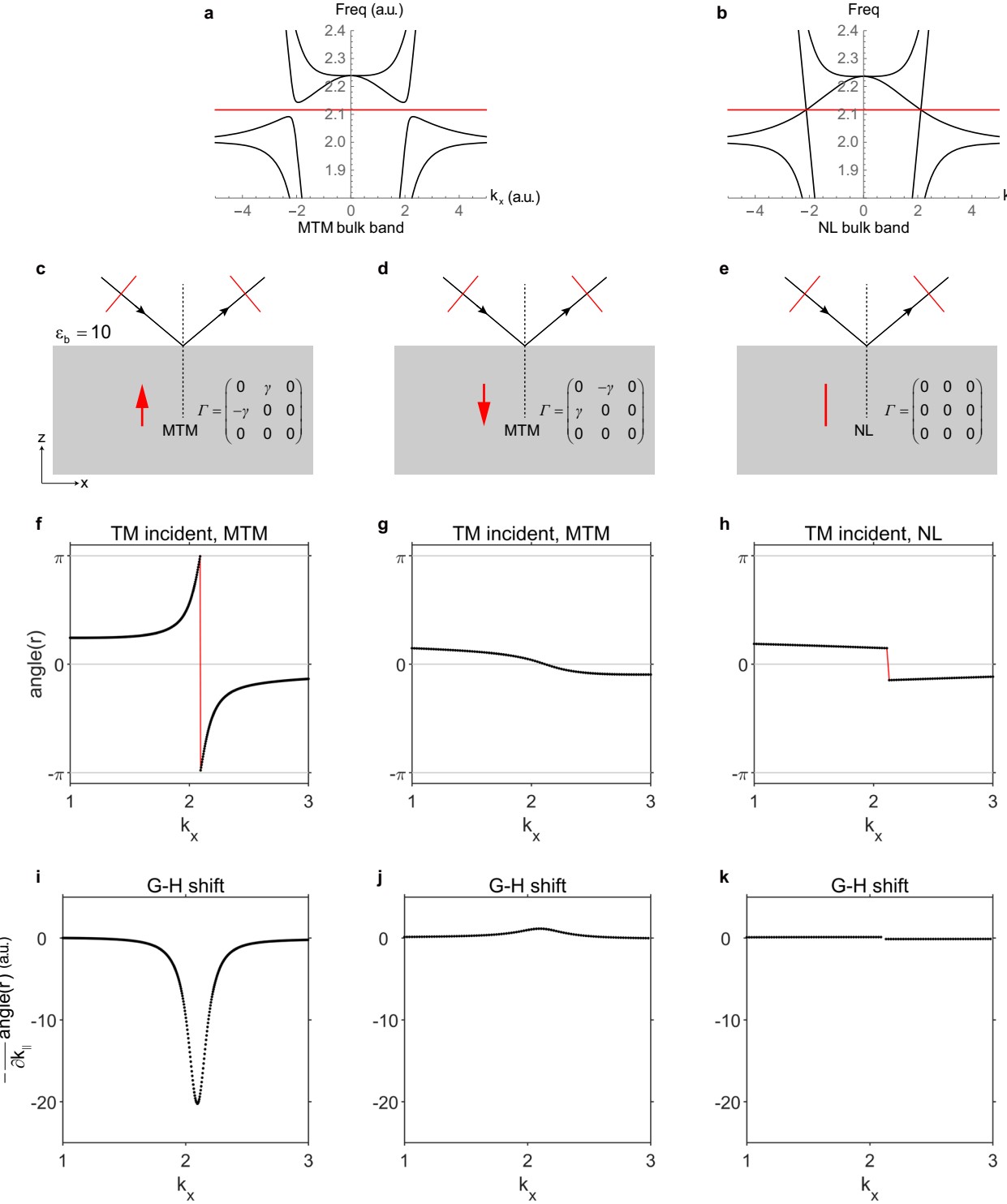

**Fig. 5 Reflection phases on the front and back surfaces of the momentum-space toroidal moment (MTM) and nodal line (NL).** **a**, **b** Bulk band structure, the red line indicates the frequency we used in the calculation. **c**–**e** The reflection configurations. The red arrows indicate the orientations of the MTM. Nodal line is non-orientable indicated with a red line segment. The incident and reflected beams are TM (transverse magnetic, $H_z = 0$) polarized as illustrated with thin red line segments. **f**–**h** Reflection phases for the corresponding three different cases with dielectric constants $\varepsilon_b = 10$. **i**–**k** The spatial shifts for the reflected beams obtained with $-\frac{\partial}{\partial k_{||}} \text{angle}(r)$.

these three cases. One sees that on the front surface the reflection phase angle($r$) increases by $2\pi - \Delta$ ($\Delta < 2\pi$), while on the back surface the reflection phase decreases by about $\Delta$. The gradient of reflection phase is linked to the anomalous shift $\ell^s = -\frac{\partial}{\partial k_{||}} \text{angle}(r)$ (including both Goos–Hänchen effect and Imbert–Fedorov effect;

note that the definition may differ from general ones with a minus sign)[43]. The different reflection phase distributions on the front and back surfaces will induce negative and (tiny) positive Goos–Hänchen shifts (Fig. 5i, j), respectively. Therefore, the Goos–Hänchen shifts with different signs on the two surfaces serve

as another important signature of MTM. In addition, the striking difference between reflection phases on the front and back surfaces shows that the MTM is orientable.

**Transverse spin**. Another interesting feature of a system with MTM is the presence of transverse spin. Supplementary Fig. 10 shows that for a given frequency closing to the bandgap (red lines in Supplementary Fig. 10a) the corresponding electric field polarization states (Supplementary Fig. 10b) are elliptically polarized. These electric fields rotate along the propagation direction in a similar way as a bicycle wheel, which is therefore called transverse spin. Although transverse spin is common for surface plasmon polaritons[44], its realization in bulk modes remains rare[45] and may bring about many interesting applications, such as transverse spin–orbital coupling and transverse spin beam shifts at an abrupt interface. In addition, because the Berry curvature distribution is related to the transverse spin, those phenomena associated to transverse spin are also linked to Berry curvature, providing another way towards characterizing Berry curvature. Supplementary Fig. 10c gives the transverse spin defined as $\mathrm{Im}\left(\mathbf{E}^* \times \mathbf{E}\right)_x$, from which we clearly see the transverse spin reaches its maximum/minimum at the valley position.

## Discussion

It has been well accepted that the positive/negative Weyl points as sources/drains of Berry curvature can emit/collect Berry flux in the 3D momentum space. Here we show that in a gapped nodal line system, the Berry flux is completely sourceless and it has the form of closed loops. For a $PT$ symmetric two-band nodal line, i.e., $H = f(k_x, k_y, k_z)\sigma_x + g(k_x, k_y, k_z)\sigma_z$, we can apply the $\mathbb{Z}$ classified topological charge to characterize it, i.e., the loop encircling the nodal line accumulates $n\pi$ ($n \in \mathbb{Z}$) Berry phase[46]. Once we add a small perturbation described by $h\sigma_y$, where we assume $h$ is a very small constant. The nodal line will be fully gapped. Then the Berry curvature distribution concentrated in a thin tube will follow the previous nodal line configuration. As nodal lines are common and easy to control, one may manipulate Berry curvature distribution via gapping the elaborately designed nodal lines.

Our work reveals that by properly breaking the degeneracy along the nodal ring, the induced Berry curvature with remaining localized/confined around the original profile of the nodal ring for a sufficiently small gap, can exhibit toroidal moment. Benefitting from the flexibility of metamaterial design for manipulating the topology of band structure, our work provides a way for investigating phenomena associated with the toroidal distribution of Berry curvature. The presence of MTM leads to various interesting phenomena including surface-dependent anomalous shifts, reconfigurable helical waveguide modes, and bulk transverse spin, etc. Moreover, the toroidal structure of Berry curvature shows non-vanishing curl[47] $\nabla \times \mathbf{\Omega}(\mathbf{k})$, corresponding to "electric" currents in the momentum space[48]. They behave as new sources for generating Berry curvature in parallel with the "magnetic" charges-Weyl points. Recently it was discovered that the Berry curvature dipole can contribute to quantum nonlinear Hall effect in condensed matter systems[49–51]. Berry curvature toroidal moments also contribute to the nonlinear topological responses[52]. Extension to more complicated nodal line configurations, such as gapping of nodal chains[46,53,54], nodal links[55,56], and nodal knots[57,58], may lead to various exotic 3D Berry curvature distributions in relation to the orientation of nodal lines and non-Abelian topological charges[46]. Our scheme may also be generalized into higher-dimension, multi-band environments such as in the spinful system with spin–orbit coupling[59].

## Methods

**Sample fabrication**. Supplementary Fig. 2a shows the sample fabricated with commercial PCB (Printed Circuit Board) technology, where face (front) and back sides are defined. There are 75-unit cells along both directions with total area of $300 \times 300$ mm$^2$, where each unit cell is $4 \times 4 \times 2$ mm$^3$. Each periodic layer consists of one structure layer (1 mm-thick) and one blank layer (1 mm-thick).

**Experimental setup**. The schematic diagram of the experimental setup for bulk state mapping is shown in Supplementary Fig. 2b. A near-field dipole antenna acts as the source (red). The dipole moment is mainly polarized along the antenna. Another near-field antenna of the same configuration with dipole polarization arbitrarily oriented acts as the probe (purple). The source and probe are connected by a microwave vector network analyser (VNA, Keysight N5234B, University of Birmingham). As the probe scans above the surface, amplitude and phase information of the electromagnetic field are collected via S-parameter which can be further Fourier transformed into the distribution in the momentum space. The scan step is set to be 3 mm along both directions, which determines the Fourier transformation range (4/3 FBZ). A frequency range of 10–18 GHz with 801 sample points is set.

Supplementary Fig. 2c shows the line scan of interface states. The source is positioned around one corner of the interface (red), which ensures that the excited modes are mainly interface states. Another probing antenna line-scans the interface with step of 3 mm and frequency range of 10–18 GHz (801 sample points).

We use the resonance dip of one structure layer transmission to retrieve the background dielectric constant. Supplementary Fig. 3a shows the schematic view of the setup. With setting the relative permittivity to be 1.8, the simulation and experiment results fit well, as shown in Supplementary Fig. 3b.

## Data availability

The data that support the findings of this study are available from the corresponding authors upon reasonable request.

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

## Acknowledgements

This work is supported by the Hong Kong RGC (AoE/P-02/12, 16304717, 16310420) and the Hong Kong Scholars Program (XJ2019007). S.Z. acknowledges support from the ERC Consolidator Grant (TOPOLOGICAL), the Royal Society, and the Wolfson Foundation. C.-X.L. acknowledges the support of the Office of Naval Research (Grant No. N00014-18-1-2793) and Kaufman New Initiative research Grant No. KA2018-98553 of the Pittsburgh Foundation.

## Author contributions

B.Y., C.T.C., C.-X.L., and S.Z. conceived the idea; B.Y. designed the sample with input from C.T.C. and S.Z.; Y.B. carried out all measurements with help from O.Y.; B.Y., C.-X.L., and S.Z. developed and carried out the data analysis; R.-X.Z., R.-Y.Z., Z.Z., J.F., and H.S. participated in the analysis and discussion of the results. C.T.C., C.-X.L., and S.Z. supervised the whole project. B.Y. wrote the manuscript with input from all other authors.

## Competing interests

The authors declare no competing interests.
