## [Peer Review File · Nature Communications]

Reviewers' Comments:

Reviewer #1:

Remarks to the Author:

The manuscript "Toroidal Moment in the Momentum Space" reports the observation of a toroidal moment of Berry curvature in a photonic crystal by opening a spectral gap in a nodal line system.

The system studied is a variation of the one characterized by some of the authors in a previous publication [Nature Commun. 9, 950 (2018)]. There, a mirror symmetry protects a topological nodal line in the spectrum of the system. Here, the very same symmetry is broken and the nodal line is gapped. The distribution of Berry curvature in momentum space emerging from this gap-opening transition is carefully investigated.

The manuscript is well structured, well-written, and the experimental work seems technically rigorous. Yet, I am unsure whether the evidence provided supports the claims made and I find the discussion of the results at times deceiving.

Let me discuss my main concerns:

1. The expression 'nearly quantized' recurs in the text. This statement is misleading. In the absence of a protecting symmetry, nothing is quantized or topological in this system. The value of the Berry curvature can be continuously tuned to zero by stronger symmetry breaking perturbations. As soon as the gap opens and the symmetry is broken, nothing prevents from completely trivializing the system. This makes the system not generic and less robust to disorder. Only a brief sentence touches on this point while clarifying it should be crucial. It becomes particularly striking when comparing the proposed toroidal moment of Berry curvature to Weyl points and their respective robustness.

2. I doubt the observed phenomenon enjoys symmetry protection and topological robustness. Still, the photonic crystal is carefully designed to open a small gap and preserves the influence of the nodal line in the distribution of the Berry curvature for the gapped system. This careful design leads to the emergence of a toroidal moment. This alone could be a note-worthy result if the presence of a toroidal moment would lead to new phenomenology. I fail to see a characterization of these novel phenomena. Similar edge states would be observed in a nodal line semimetal (albeit not at momenta where the bulk is gapless). Moreover, in the latter, they would be topologically protected, while I would expect the edge states here reported to disappear or gap-out for increasing bulk-gap size. Is the proposed design of interface with arbitrary refraction index, i.e., negative and positive, also shared with nodal line semimetals? If not and entirely due to the toroidal moment, it is a pity that this phenomenon is not experimentally studied.

Here a list of other minor comments and requests for clarification:

- The blue circle in Fig. 1a confuses me. That circle is in the energy-momentum plane rather than in the k_y - k_z plane as stated in the text.
- What enforces the toroidal structure? Other perturbations would lead to different gap-opening transitions, e.g. time-reversal breaking would lead to Weyl points. Would any perturbations that fully gaps the bulk lead to a toroidal moment, for sufficiently small bulk-gap? Is it somehow protected by rotation invariance around the z -axis?
- Why is the Wilson loop operator computed along the base loop Γ -X-M- Γ rather than along a reciprocal lattice vector? Loops along high-symmetry lines might be relevant for crystalline symmetries which do not seem to play a role in the physics discussed here.
- Are the dashed line in Fig. 3 b-e-h associate to modes in the air domain? It would be helpful if this is clearly stated also in the figure's caption.
- What is the scale of the colormap in Fig. 2i? How would that 0.1 compare to the case of a nodal line semimetal?

In conclusion, I do not consider the results and supporting experimental evidence in the

manuscript sufficient to meet the criteria of Nature Communications. Some findings are potentially interesting, e.g., the possibility to realize a negative and positive refraction index at the interface, but their robustness and relevance for future works are not discussed.

Reviewer 2:

In this manuscript, the authors report on the demonstration of a momentum-space toroidal moment in a photonic metamaterial. The corresponding toroidal Berry curvature is obtained by starting from a known nodal ring phase, breaking its protecting mirror symmetry (M_z) and, consequently, fully gapping the spectrum. Depending on the size of the newly arisen gap, the strength of the magnetic toroidal moment varies and the flux of the corresponding Berry curvature deviates from the exactly quantized values ($\pm\pi$). As a consequence, there arise gapless helical states at the interfaces of metamaterials with opposite orientations. These theoretical predictions are confirmed in a photonic metamaterial fabricated by the printed circuit board technology.

The manuscript starts with an introduction on Berry phase considerations in materials and metamaterials, and the motivation behind the quest for the realization of magnetic toroidal moments. After the specific model with the line nodal ring phase is introduced, its underlying topological properties are theoretically explained and the path towards mirror symmetry breaking introduced. The band structure is calculated, for a later comparison with experimental results. Wilson loop calculations show the two-dimensional Berry curvature distribution with sinks and sources in the plane that cuts orthogonally through the ring position in momentum space, as well as the approximate quantization to 0 or $\pm\pi$ of the Wannier centers.

The model is realized in a photonic metamaterial, whose mirror symmetry is broken by introducing additional small metallic bars in the system. Bulk states are measured by near-field scans on the interface between the metamaterial and air, while the predicted gapless helical states by line-scanning at the interface of two oppositely oriented metamaterials. The extracted dispersions for both the bulk and the surface states are shown to agree rather well with the theoretical predictions.

The key contributions of the presented manuscript are as follows:

- Theoretical prediction and experimental realization of a photonic metamaterial with non-vanishing magnetic toroidal moment in momentum space.
- Theoretical investigations of the properties and consequences of the non-vanishing toroidal moment and quasi-quantized Berry fluxes.
- Experimental detection of the gapless helical states at the interfaces of oppositely oriented metamaterials.

The results presented in the manuscript are interesting, scientifically sound and worth publishing. I am not, however, certain that the Nature Communications, which aims to report on important advances relevant for a broad audience, is the appropriate journal for the publication. Before the final recommendation, there are several issues that need to be clarified:

- One of the most relevant properties of topological matter is the robustness of its phenomena, due to the quantization of topological charges that cannot be changed by smooth local changes. Throughout the manuscript, the authors emphasize more than once the “nearly quantized Berry curvature fluxes”. What exactly is the importance of such non-quantized Berry fluxes? If the robustness is gone, once in the non-vanishing magnetic toroidal moment regime, are the advantages usually related to topological matter also gone?
- How exactly do the interesting concepts and applications earlier developed for electromagnetic materials with toroidal magnetic moment translate in the case of the realization of such moments in momentum space? Besides just being a new form of Berry curvature, what kind of consequences would it have on e.g. the mentioned nonlinear topological responses?
- How are the field distributions in Fig. 4 obtained? Are they only theoretical or also experimental?
- Why exactly do the authors state that they provide a “new mechanism for the formation of Berry curvature”? It is not clear to me what “new mechanism” is supposed to mean. Isn’t this just a new form/ 3D field of the Berry curvature (although possibly interesting)?
- In line 135, the authors mention a slight frequency shift of the experimental results as compared to theoretical considerations. This is, however, not justified. What is the reason for such a disagreement?
- There are several typos to be corrected, as well as certain formulations to be improved. I would highly suggest to find a native English speaker to read the manuscript once corrected.
 - e.g. 26 “exhibit” → “exhibits”
 - 57 “physic” → “physics”
 - 153 “re-survived” ??
 - 159 “have possess transverse spin” ??

Reviewer #3:

Remarks to the Author:

The current work by B. Yang et al., discusses the formation of momentum-space toroidal moments, forming a closed-loop Berry flux. This idea is neat and interesting and can open up new directions in „topological photonics“. I have though a few major comments that the authors should consider to expand the discussions and describing better the novelty and new physics of their work.

1. My main concern is that the formation of looping Berry flux does not seem to be new. As the authors describe, spin-orbit coupling happening concomitant with band inversion – in its simplest form – might lead to either Weyl points or gaped band structure. The former causes Weyl semimetals and the latter topological insulators that at their surfaces, they support distinguished electronic (or photonic) surface states (see for example the review B. Yan and C. Felser, *Annu. Rev. Condens. Matter Phys.* 2017. 8:1-19). The former as well supports “magnetic monopoles” and the latter “magnetic loops”, that one can call them a toroidal moment in the momentum space. Can only naming this feature as a toroidal moment lead to a novel discovery?

2. In connection with the above comment, what did we learn about this toroidal moment? The fact that gaped topological insulators can support surface waves is not a new finding. Does this toroidal moment aspect lead to new physics? What kind of new features for surface waves were observed here that has yet not been reported in the literature?

3. The authors discuss that the reciprocal toroidal moment has different time-space symmetry rules. What is the effect of this aspect on the surface states or bulk states? An experimental proof for this claim can indeed unambiguously support the existence of the toroidal moment.

4. There are certain disagreements between theory and experiment. For example, it seems that in the experimental results one of the surface states is missing (Fig. 3f and i). What is the reason for this asymmetry in the band structure?

In general, although I have found the results beautiful and overall convincing, much more explanations about the novelty and physics of the observations here are required.

Reviewer #1 (Remarks to the Author):

Comment 1.1: The manuscript "Toroidal Moment in the Momentum Space" reports the observation of a toroidal moment of Berry curvature in a photonic crystal by opening a spectral gap in a nodal line system.

The system studied is a variation of the one characterized by some of the authors in a previous publication [Nature Commun. 9, 950 (2018)]. There, a mirror symmetry protects a topological nodal line in the spectrum of the system. Here, the very same symmetry is broken and the nodal line is gapped. The distribution of Berry curvature in momentum space emerging from this gap-opening transition is carefully investigated.

Response 1.1: We thank the referee for the careful reading and the nice summary of our work. We also thank the referee for the following critical comments to improve the presentation of our work.

Comment 1.2: The manuscript is well structured, well-written, and the experimental work seems technically rigorous.

Response 1.2: We thank the referee for his/her recognitions and positive assessments of our work.

Comment 1.3: Yet, I am unsure whether the evidence provided supports the claims made and I find the discussion of the results at times deceiving.

Response 1.3: Our paper shows the way to realize momentum space toroidal moment, where Berry curvature exhibits a vortex-like configuration without any source and drain in the momentum space. We start from a mirror-symmetry protected nodal ring semimetal and then gap it via breaking the mirror symmetry to realize the momentum space toroidal moment. We first numerically calculated Berry phase (Wannier centre) and Berry curvature to theoretically verify our proposal. We found that a key signature of the toroidal Berry curvature configuration is the presence of helical domain wall states. Then we carry out the near-field scanning measurement to map out both the bulk and the domain wall states. The experimental result agrees well with our simulation. Therefore, both theoretical and experimental evidences support our claim - the system exhibits momentum space toroidal moment.

We apologize for any confusion in the text. In the revised manuscript, we have tried to make the discussions clearer.

Let me discuss my main concerns:

Comment 1.4: 1. The expression 'nearly quantized' recurs in the text. This statement is misleading. In the absence of a protecting symmetry, nothing is quantized or topological in this system. The value of the Berry curvature can be continuously tuned to zero by stronger

symmetry breaking perturbations. As soon as the gap opens and the symmetry is broken, nothing prevents from completely trivializing the system. This makes the system not generic and less robust to disorder. Only a brief sentence touches on this point while clarifying it should be crucial. It becomes particularly striking when comparing the proposed toroidal moment of Berry curvature to Weyl points and their respective robustness.

Response 1.4: We thank the referee for pointing this out.

In the text, “nearly quantized” means that the Berry curvature is quantized in the limit of very small (diminishing) bandgap. While we admit the terminology “nearly quantized” maybe confusing, the frequently used notion of “valley Chern number” is based on exactly the same principle. Although it is usually less emphasized, the valley Chern number is also just “nearly quantized” and the quantization is exact only in the limit of diminishing bandgap and the separation of different valleys. So far, the concepts of quantum valley Hall effect and valley Chern number have been well accepted. Especially in photonics/phononics, researchers have proposed various potential applications based on the valley physics, such as topological laser¹, communications², waveguides³, etc. The quantum valley Hall effects are robust as long as the gap is small enough^{4,5}. In fact, the less symmetry requirement makes valley related effects much more accessible in experimental implementation for different types of waves, which is an important advantage compared with those topological phases that are strictly symmetry protected. The toroidal moment demonstrated here represents a 3D generalized ring shaped valley.

To elucidate our argument, in the revised manuscript we have changed the phrase “nearly quantized” to be “quasi-quantized” as suggested by the 2nd reviewer, and added the following paragraph in page 4.

“It is worth noting that for each vertical cutting plane (planes that contain k_z axis), e.g., the $k_y - k_z$ plane, the topological features can be well captured by valley Chern number. Note that the valley Chern number is quasi-quantized and the quantization is exact only in the limit of diminishing bandgap. In fact, the less symmetry requirement makes valley related effects much more accessible in experimental implementation and for the universality of all kinds of waves, which is an important advantage compared with those topological phases strictly protected by symmetries. So far, 2D valley physics has promised a plethora of applications. Especially in photonics/phononics, researchers have proposed topological laser¹, communications², waveguides³, etc. The valley effects are very robust as long as the gap is small enough^{4,5}.”

Comment 1.5: 2. I doubt the observed phenomenon enjoys symmetry protection and topological robustness. Still, the photonic crystal is carefully designed to open a small gap and preserves the influence of the nodal line in the distribution of the Berry curvature for the gapped system. This careful design leads to the emergence of a toroidal moment. This alone could be a note-worthy result if the presence of a toroidal moment would lead to new phenomenology. I fail to see a characterization of these novel phenomena. Similar edge states would be observed in a nodal line semimetal (albeit not at momenta where the bulk is gapless). Moreover, in the latter, they would be topologically protected, while I would expect the edge states here reported to disappear or gap-out for increasing bulk-gap size. Is the proposed design of interface with arbitrary refraction

index, i.e., negative and positive, also shared with nodal line semimetals? If not and entirely due to the toroidal moment, it is a pity that this phenomenon is not experimentally studied.

Response 1.5: Our toroidal moment system with a small band-gap introduced by mirror symmetry breaking possesses dramatically different edge states from that of the nodal line system without the symmetry breaking. Nodal line system may support drumhead surface states (but not guaranteed) which can be made perfectly flat with zero group velocity and thus not helical. The usual argument about the symmetry protection of the drumhead surface states in a nodal line system is quite ambiguous (except for the case protected by chiral symmetry belonging to the AIII class) because there is no mirror symmetry on the surface, although the nodal line is protected by mirror symmetry of the bulk medium. Drawing the argument based on topological crystalline insulators, symmetry protected topological surface states only exist on the surface which shares the same symmetry with their bulk states. In addition, π Berry phase is not a sufficient condition to guarantee the presence of the stable surface states. Their presence crucially depends on the termination and the crystal structure in the unit cell⁶.

With the system possessing toroidal moment, the interface states are helical (chiral for each valley) across the complete bandgap and can be well explained by valley Chern number for each 2D cutting plane, e.g., $k_x/k_y - k_z$ plane. As long as the gap is small enough, the helical edge states are robust and cannot be removed.

Following the reviewer's suggestion, we carry out a surface state simulation for the nodal line configuration as shown in Fig. R1. There are no surface states for both PEC (perfect electric conductor) and PMC (perfect magnetic conductor) boundary conditions (The two boundary conditions are very common in photonics and are analogous to hard boundaries in condensed matters). Due to finite size effect of the supercell, there is a small gap as illustrated by solid lines in Fig. R1a. The dashed lines indicate nodal lines calculated with a single unit cell with periodic boundary conditions along all x, y and z directions.

Fig. R1. (Fig. S8 in supplementary information) No surface states for the nodal line configuration with both PEC (perfect electric conductor) and PMC (perfect magnetic conductor)

boundary conditions (BC). The solid lines indicate the bands calculated with the supercell shown in (b). There is a small gap in the middle due to the finite size effect of the supercell. Dashed lines are the bands of nodal lines calculated by imposing periodic boundary conditions along all x, y and z directions on a single unit cell (inset in (a)).

We perform a similar simulation as shown in Fig. R2 for the momentum toroidal moment. We find when setting the front and back surfaces to be PEC and PMC boundary conditions, respectively, the front and back surfaces support helical surface states. Here the PEC/PMC boundary condition serving as mirror ($M = -1/+1$) provides the other half Chern number⁷. This finding shows the vast difference between the system with momentum toroidal moment and that with nodal line.

Fig. R2. (Fig. S9 in supplementary information) Surface states (blue and red) supported by momentum toroidal moment with PEC (perfect electric conductor) and PMC (perfect magnetic conductor) boundary conditions on the front and back surfaces, respectively. a, Calculated surface states. b, Supercell with PEC/PMC boundary condition (BC) imposed on the front/back surface. c, The corresponding surface states (SS) as marked in (a).

In order to further highlight the novel topological features of momentum toroidal moment, in the revised manuscript, we have added a new section in the supplementary materials as “**Supplementary Information IV: Surface states with PEC/PMC boundary conditions**”, where the above two figures, Fig. R1 and R2, are involved.

Besides the observed positive and negative-index interface states, we further propose the following interesting phenomena originating from the presence of momentum toroidal moment.

New phenomena related to momentum toroidal moment

1. Different anomalous beam shifts on the front and back surfaces

We calculate the reflection phase on both the front and back surfaces from effective media theory. The configurations are shown in Fig. R3b. We also present the reflection phase for the nodal line system as a reference. The reflection amplitudes are all unity as the selected frequency is located in the gap or right at the nodal degeneracy (Fig. R3a), whereas only the phase of reflection change with the in-plane wavevector k_x . A TM (transverse magnetic $H_z = 0$, indicated as red lines in (b)) beam is incident onto the front/back surface. Fig. R3c shows the dependence of reflection phases over the incident angle for the three cases. One sees that on the front surface the reflection phase $\arg(r)$ increases by $2\pi - \Delta$ ($\Delta < 2\pi$), while on the back surface the reflection phase decreases about Δ . The gradient of reflection phase is linked to the anomalous shift $\ell^s = -\frac{\partial}{\partial k_{\parallel}} \arg(r)$ (Goos-Hänchen effect)⁸. The different reflection phase distributions for the front/back surface will induce either negative or positive Goos-Hänchen shift (Fig. R3d). The Goos-Hänchen shifts of different signs in the two opposite surfaces serves as another important signature of momentum toroidal moment. In addition, the striking difference of reflection phases on the front and back surfaces shows that the momentum toroidal moment is orientable. Furthermore, the difference does not depend on local surface structures (although the front and back surfaces in realistic structures have different patterns). In other words, it is one of bulk properties of the momentum toroidal moment.

Fig. R3. (Fig. 5 in the main text) Reflection phases on the front and back surfaces of the momentum toroidal moment (MTM) and nodal line (NL). a, Bulk band structure, the red line indicates the frequency we used in the calculation. b, The reflection configurations. The red arrows indicate the orientations of the MTM. Nodal line is non-orientable indicated with red line segment. The incident and reflected beams are TM (transverse magnetic, $H_z = 0$) polarized as illustrated with thin red lines. c, Reflection phases for the corresponding three different cases. d, The spatial shift for the reflected beam obtained with $-\frac{\partial}{\partial k_{\parallel}} \arg(r)$.

2. *Transverse spin in the momentum toroidal moment*

Another interesting feature of a system with momentum toroidal moment is the presence of transverse spin. Fig. R4 shows that for a given frequency close to the bandgap (red lines in Fig. R4a) the corresponding electric field polarization states (Fig. R4b) are elliptically polarized. These electric fields rotate along the propagation direction in a similar way as a bicycle wheel, which is therefore called transverse spin. Although transverse spin is common in surfer plasmon polaritons⁹, its realization in bulk modes remains rare¹⁰ and may lead to many interesting applications, such as transverse spin-orbital coupling, transverse spin beam shift at an abrupt interface. In addition, because the Berry curvature distribution is related to the transverse spin, those phenomena associated to transverse spin is linked to Berry curvature, therefore providing another way for characterizing the Berry curvature. Figure R4c gives the transverse spin defined as $Im(\vec{E}^* \times \vec{E})$, from which we clearly see the transverse spin reaches its maximum at the valley positions.

Fig. R4. (Fig. S10 in supplementary information) Transverse spin in the momentum toroidal moment (MTM). a, The bulk band structure of MTM and nodal lines (NL). The red lines indicate the frequency used to plot the eigen electric fields. b, Elliptically and linearly polarized electric states in MTM and NL correspondingly. In the MTM the elliptical electric field orbits induce transverse spins pointing in the plane. c, Transverse spin defined as $\text{Im}(\vec{E}^* \times \vec{E})_x$, calculated with the 2nd band for both MTM and NL.

In the revised manuscript, we have added Fig. R3 (Fig. 5) and the following discussions (on page 9) into the main text to further stress the importance of momentum toroidal moment.

“Finally, we calculate the reflection phase on both the front and back surfaces based on effective media theory (see parameters in SI. V). The configurations are shown in Fig. 5. We also present the reflection phase for the nodal line system as a reference. The reflection amplitudes are all unity as the selected frequency is located in the gap or right at the nodal degeneracy (Fig. 5a), whereas only the phases of reflection change with the in-plane wavevector k_x . A TM (transverse magnetic $H_z = 0$, indicated as red lines in (b)) beam is incident onto the front/back surface. Figure 5c shows the dependence of reflection phases over the in-plane wavevector for these three cases. One sees that on the front surface the reflection phase $\arg(r)$ increases by $2\pi - \Delta$ ($\Delta < 2\pi$), while on the back surface the reflection phase decreases by about Δ . The gradient of reflection phase is linked to the anomalous shift $\ell^s = -\frac{\partial}{\partial k_{\parallel}} \arg(r)$ (including Goos-Hänchen effect and Imbert-Fedorov effect)⁸. The different reflection phase distributions for the front and back surfaces will induce either negative or (tiny) positive Goos-Hänchen shift (Fig. 5d). Therefore, the Goos-Hänchen shifts with different signs on the two surfaces serve as another important signature of MTM. In addition, the striking difference of reflection phases on the front and back surfaces shows that the MTM is orientable.

Another interesting feature of a system with momentum toroidal moment is the presence of transverse spin. Fig. S10 shows that for a given frequency close to the bandgap (red lines in Fig. S10a) the corresponding electric field polarization states (Fig. S10b) are elliptically polarized. These electric fields rotate along the propagation direction in a similar way as a bicycle wheel, which is therefore called transverse spin. Although transverse spin is common for surface plasmon polaritons⁹, its realization in bulk modes remains rare¹⁰ and may bring about many interesting applications, such as transverse spin-orbital coupling, transverse spin beam shift at an abrupt interface. In addition, because the Berry curvature distribution is related to the transverse spin, those phenomena associated to transverse spin is also lined to Berry curvature, providing another way towards characterizing Berry curvature. Figure S10c gives the transverse spin defined as $Im(\vec{E}^* \times \vec{E})$, from which we clearly see the transverse spin reaches its maximum at the valley positions.”

We moved Fig. R4 to the supplementary materials.

In addition, the proposed positive/negative refractive index cannot be realized in nodal line semimetals. In nodal line semimetals the refraction index of drumhead surface states can be zero (exactly flat), positive or negative depending on the surrounding medium. On the contrary, the interface states of momentum toroidal moment exhibit either positive (F-F interface in Fig. 3g) or negative (B-B interface in Fig. 3d) refractive indices depending on the interface configurations. Based on the valley Chern number argument the phenomena arise from the special Berry curvature distribution, namely momentum toroidal moment. Our experiments (Fig. 3) clearly demonstrate the positive/negative indices of interface states. The experiment results fit well with both the theoretical analysis and numerical simulation. The positive/negative indices associated with the interface modes are illustrated in Fig. 4.

In order to further clearly show the negative refraction phenomena, in the revision turn, we have performed a lensing experiment and added the following sentence in the main text (on page 8),

“To clearly demonstrate the two-dimensional negative refraction, we experimentally construct a Veselago lens configuration¹¹ and show the ability of focusing light from a point source to a focal point in Fig. S7.”

Fig. R5. (Fig. S7 in supplementary information) Negative refraction of two-dimensional interface states. a, Configuration used to characterize the negative refraction phenomena. The Face-Face/Back-Back supporting positive/negative refraction index is located on left/right hand side. The source antenna is located on the leftmost, two different probe antennas are used to scan the interface (or guiding) waves. The copper foil is used to break mirror symmetry on the interface and thus Face-Face interface states can excite Back-Back interface states across the sharp domain wall ($x=45$ mm in (b)). c, Negative refraction phenomena at different frequencies. Here the negative refraction effect enables perfect lens, i.e., bringing the light from a point source to a focal point on the right hand side.

Here a list of other minor comments and requests for clarification:

Comment 1.6:- The blue circle in Fig. 1a confuses me. That circle is in the energy-momentum plane rather than in the k_y - k_z plane as stated in the text.

Response 1.6: We apologize for the confusion.

In the revised manuscript, we have added an inset in Fig. 1 to make the description clearer. It is shown below as well (Fig. R6).

Fig. R6. (Fig. 1 in the main text) Monopoles and toroidal moments in the momentum space.

Comment 1.7:- What enforces the toroidal structure? Other perturbations would lead to different gap-opening transitions, e.g. time-reversal breaking would lead to Weyl points. Would any perturbations that fully gaps the bulk lead to a toroidal moment, for sufficiently small bulk-gap? Is it somehow protected by rotation invariance around the z -axis?

Response 1.7: We thank the referee for the valuable questions.

Yes, different perturbations may lead to different topological phases. The transition from nodal line to Weyl points has been well understood and gapping a nodal line may also lead to a topological insulator¹². Thus, not all of perturbations that fully gap the nodal line will result in a momentum toroidal moment.

Because the existence of nodal ring is not protected by the rotation invariance around the z -axis, the momentum toroidal moment resulting from breaking the mirror symmetry of the nodal line system is not restricted by this constraint either. Since the Berry curvature of the MTM forms a vortex-like configuration, breaking C_4 rotation will only distort but cannot destroy this configuration. Thus the C_4 rotation symmetry around z -axis in our work is not a necessary condition.

In the revised supplementary information in have added the below discussion on page 6.

“Here the momentum toroidal moment arises from gapping of a nodal ring. Usually for sufficiently small band-gap, the Berry curvature is tightly concentrated around a ring to form the momentum toroidal moment. As the nodal ring does not require the protection of rotation invariance around the z -axis, the momentum toroidal moment resulting from breaking the mirror symmetry of the nodal line system is not restricted by this constraint

either. Thus the C_4 rotation symmetry around z-axis in our work is not a necessary condition.”

Comment 1.8:- Why is the Wilson loop operator computed along the base loop Gamma-X-M-Gamma rather than along a reciprocal lattice vector? Loops along high-symmetry lines might be relevant for crystalline symmetries which do not seem to play a role in the physics discussed here.

Response 1.8: Because the meta-crystal we designed possesses C_{4v} symmetry around z axis, the path $\Gamma - X - M - \Gamma$ is helpful to explicitly show the Wannier centre. Once the Wannier centre along the high-symmetry line is calculated, other general paths can be predicted accordingly since Wannier centre can only evolve smoothly. Furthermore, the C_{4v} symmetry here also renders the Berry curvature distribution nearly circular. The appearance of momentum toroidal moment does not rely on C_{4v} , but its existence makes the Berry curvature configuration more symmetrical.

Comment 1.9:- Are the dashed line in Fig. 3 b-e-h associate to modes in the air domain? It would be helpful if this is clearly stated also in the figure's caption.

Response 1.9: Yes, the dashed lines in Fig. 3b,e,h correspond to modes in the air domain. We have updated it in the revised manuscript (in the caption of Fig. 3).

Comment 1.10:- What is the scale of the colormap in Fig. 2i? How would that 0.1 compare to the case of a nodal line semimetal?

Response 1.10: Figure. 2i shows the Berry phase calculated in each small square loop as defined in Fig. S1. Its distribution in the momentum space correspond to Berry curvature. Thus the scale of the colormap is the Berry phase in unit of rad. Although Berry curvature and Berry phase have different units, here we still can use the distribution of Berry phase to indicate that of Berry curvature.

For the nodal line semimetal, Berry curvature is expressed by a delta function as $\Omega = \delta(\vec{k} - \vec{k}_{NL})$. In other words, the nodal line system has both time-reversal and inversion symmetries, the Berry curvature is vanishing everywhere in the momentum space except it becomes singular on the nodal line.

Comment 1.11: In conclusion, I do not consider the results and supporting experimental evidence in the manuscript sufficient to meet the criteria of Nature Communications. Some findings are potentially interesting, e.g., the possibility to realize a negative and positive refraction index at the interface, but their robustness and relevance for future works are not discussed.

Response 1.11: In the paper we propose momentum space toroidal moment that possesses Berry vortex-like curvature distribution and supports helical interface states, and we illustrate the corresponding physical consequence. We experimentally map out the bulk and interface states

which agree well with the simulation and theoretical expectation. Moreover, our experiment explicitly shows the positive and negative refraction indices of interface states.

The momentum toroidal moment we propose will introduce many interesting phenomena, including anomalous beam shifts of opposite signs on the front and back surfaces. The MTM also offers a platform for study of transverse spin of bulk modes, which support many intriguing effects including transverse spin-orbital coupling, transverse spin induced beam shift at the interface, etc.

To emphasize this, in the revised manuscript we have added the above mentioned phenomena originating from the momentum toroidal moment. In the conclusion section, we further summarize the potential observables of momentum toroidal moment as shown below.

“The presence of MTM leads to various interesting phenomena including surface-dependent anomalous shift, reconfigurable helical waveguide modes and bulk transverse spin, etc.”

Reviewer #2 (Remarks to the Author):

Comment 2.1: In this manuscript, the authors report on the demonstration of a momentum-space toroidal moment in a photonic metamaterial. The corresponding toroidal Berry curvature is obtained by starting from a known nodal ring phase, breaking its protecting mirror symmetry (M_z) and, consequently, fully gapping the spectrum. Depending on the size of the newly arisen gap, the strength of the magnetic toroidal moment varies and the flux of the corresponding Berry curvature deviates from the exactly quantized values ($\pm\pi$). As a consequence, there arise gapless helical states at the interfaces of metamaterials with opposite orientations. These theoretical predictions are confirmed in a photonic metamaterial fabricated by the printed circuit board technology.

The manuscript starts with an introduction on Berry phase considerations in materials and metamaterials, and the motivation behind the quest for the realization of magnetic toroidal moments. After the specific model with the line nodal ring phase is introduced, its underlying topological properties are theoretically explained and the path towards mirror symmetry breaking introduced. The band structure is calculated, for a later comparison with experimental results. Wilson loop calculations show the two-dimensional Berry curvature distribution with sinks and sources in the plane that cuts orthogonally through the ring position in momentum space, as well as the approximate quantization to 0 or $\pm\pi$ of the Wannier centers.

The model is realized in a photonic metamaterial, whose mirror symmetry is broken by introducing additional small metallic bars in the system. Bulk states are measured by nearfield scans on the interface between the metamaterial and air, while the predicted gapless helical states by line-scanning at the interface of two oppositely oriented metamaterials. The extracted dispersions for both the bulk and the surface states are shown to agree rather well with the theoretical predictions.

The key contributions of the presented manuscript are as follows:

- Theoretical prediction and experimental realization of a photonic metamaterial with non-vanishing magnetic toroidal moment in momentum space.
- Theoretical investigations of the properties and consequences of the non-vanishing toroidal moment and quasi-quantized Berry fluxes.
- Experimental detection of the gapless helical states at the interfaces of oppositely oriented metamaterials.

Response 2.1: We thank the referee for the very careful reading and the nice summary of our work.

Comment 2.2: The results presented in the manuscript are interesting, scientifically sound and worth publishing. I am not, however, certain that the Nature Communications, which aims to report on important advances relevant for a broad audience, is the appropriate journal for the publication. Before the final recommendation, there are several issues that need to be clarified:

Response 2.2: We thank the referee for his/her recognitions and positive assessments of our work.

Comment 2.3: • One of the most relevant properties of topological matter is the robustness of its phenomena, due to the quantization of topological charges that cannot be changed by smooth local changes. Throughout the manuscript, the authors emphasize more than once the “nearly quantized Berry curvature fluxes”. What exactly is the importance of such non-quantized Berry fluxes? If the robustness is gone, once in the non-vanishing magnetic toroidal moment regime, are the advantages usually related to topological matter also gone?

Response 2.3: We thank the referee for the considerate and valuable question. The terminology “nearly quantized” maybe misleading as mentioned by the 1st reviewer. The phrase “quasi-quantized Berry fluxes” as you suggested in **Comment 2.1** looks more appropriate.

The quasi-quantized Berry fluxes plays a similar role as the frequently used notion of “valley Chern number”. Although it is usually less emphasized, the valley Chern number is also just “nearly quantized” and the quantization is exact only in the limit of diminishing bandgap. So far, the concepts of quantum valley Hall effect and valley Chern number have been well accepted. Especially in photonics/phononics, researchers have proposed various potential applications based on the valley physics, such as topological laser¹, communications², waveguides³, etc. The quantum valley Hall effects are robust as long as the gap is small enough^{4,5}. In fact, the less symmetry requirement makes valley related effects much more accessible in experimental implementation for different types of waves, which is an important advantage compared with those topological phases that are strictly symmetry protected.

Thus, the quasi-quantized Berry fluxes with a small gap will guarantee gapless helical interface states. If the Berry flux deviates significantly from π with a relative large gap, the momentum space toroidal moment should still exist (Actually one can control the strength of momentum toroidal moment via gap size). However, the interface states will become gapped. Other interesting phenomena including surface-dependent anomalous beam shift and transverse spin should still survive.

Figure R7 shows the distribution of Berry curvature against the gap size γ . When γ increases, Berry curvature spreads out.

Fig. R7. Berry curvature distribution with γ and k_x when $k_y = k_z = 0$.

To elucidate our argument, in the revised manuscript we have added the following paragraph in page 4.

“It is worth noting that for each vertical cutting plane (planes that contain k_z axis), e.g., the $k_y - k_z$ plane, the topological features can be well captured by valley Chern number. Note that the valley Chern number is quasi-quantized and the quantization is exact only in the limit of diminishing bandgap. In fact, the less symmetry requirement makes valley related effects much more accessible in experimental implementation and for the universality of all kinds of waves, which is an important advantage compared with those topological phases strictly protected by symmetries. So far, 2D valley physics has promised a plethora of applications. Especially in photonics/phononics, researchers have proposed topological laser¹, communications², waveguides³, etc. The valley effects are very robust as long as the gap is small enough^{4,5}.”

Comment 2.4: • How exactly do the interesting concepts and applications earlier developed for electromagnetic materials with toroidal magnetic moment translate in the case of the realization of such moments in momentum space? Besides just being a new form of Berry curvature, what kind of consequences would it have on e.g. the mentioned nonlinear topological responses?

Response 2.4: We thank the referee for valuable question.

The mentioned nonlinear topological responses in gapped nodal line system have been analysed in Ref. ¹³ (Parity anomaly in the nonlinear response of nodal-line semimetals) in details, we also have cited the Ref. in the main text. Basically, if the nodal loop is not tilted, parity anomaly

appears in these nonlinear Hall responses for finite values of the polar angle characterizing the nodal loop, but vanishes after integrating over this polar angle.¹³

In the revision turn, we also found several new phenomena related to the momentum toroidal moment as clarified below.

New phenomena related to momentum toroidal moment

1. Different anomalous beam shifts on the front and back surfaces

We calculate the reflection phase on both the front and back surfaces from effective media theory. The configurations are shown in Fig. R3b. We also present the reflection phase for the nodal line system as a reference. The reflection amplitudes are all unity as the selected frequency is located in the gap or right at the nodal degeneracy (Fig. R3a), whereas only the phase of reflection change with in-plane wavevector k_x . A TM (transverse magnetic $H_z = 0$, indicated as red lines in (b)) beam is incident onto the front/back surface. Fig. R3c shows the dependence of reflection phases over the incident angle for the three cases. One sees that on the front surface the reflection phase $\arg(r)$ increases by $2\pi - \Delta$ ($\Delta < 2\pi$), while on the back surface the reflection phase decreases about Δ . The gradient of reflection phase is linked to the anomalous shift $\ell^s = -\frac{\partial}{\partial k_{\parallel}} \arg(r)$ (Goos-Hänchen effect)⁸. The different reflection phase distributions for the front/back surface will induce either positive or negative Goos-Hänchen shift. The Goos-Hänchen shifts of different signs in the two opposite surfaces serves as another important signature of momentum toroidal moment. In addition, the striking difference of reflection phases on the front and back surfaces shows that the momentum toroidal moment is orientable. Furthermore, the difference does not depend on local surface structures (although the front and back surfaces in realistic structures have different patterns). In other words, it is one of bulk properties of the momentum toroidal moment.

Fig. R3. (Fig. 5 in the main text) Reflection phases on the front and back surfaces of the momentum toroidal moment (MTM) and nodal line (NL). a, Bulk band structure, the red line indicates the frequency we used in the calculation. b, The reflection configurations. The red arrows indicate the orientations of the MTM. Nodal line is non-orientable indicated with red line segment. The incident and reflected beams are TM (transverse magnetic, $H_z = 0$) polarized as illustrated with thin red lines. c, Reflection phases for the corresponding three different cases. d, The spatial shift for the reflected beam obtained with $-\frac{\partial}{\partial k_{\parallel}} \arg(r)$.

2. *Transverse spin in the momentum toroidal moment*

Another interesting feature of a system with momentum toroidal moment is the presence of transverse spin. Fig. R4 shows that for a given frequency close to the bandgap (red lines in Fig. R4a) the corresponding electric field polarization states (Fig. R4b) are elliptically polarized. These electric fields rotate along the propagation direction in a similar way as a bicycle wheel, which is therefore called transverse spin. Although transverse spin is common in surface plasmon polaritons ⁹, its realization in bulk modes remains rare ¹⁰ and may lead to many interesting applications, such as transverse spin-orbital coupling, transverse spin beam shift at an abrupt interface. In addition, because the Berry curvature distribution is related to the transverse spin, those phenomena associated to transverse spin is linked to Berry curvature, therefore providing another way for characterizing the Berry curvature. Figure R4c gives the transverse spin defined as $Im(\vec{E}^* \times \vec{E})$, from which we clearly see the transverse spin reaches its maximum at the valley positions.

Fig. R4. (Fig. S10 in supplementary information) Transverse spin in the momentum toroidal moment (MTM). a, The bulk band structure of MTM and nodal lines (NL). The red lines indicate the frequency used to plot the eigen electric fields. b, Elliptically and linearly polarized electric states in MTM and NL correspondingly. In the MTM the elliptical electric field orbits induce transverse spins pointing in the plane. c, Transverse spin defined as $Im(\vec{E}^* \times \vec{E})_x$, calculated with the 2nd band for both MTM and NL.

In the revised manuscript, we have added Fig. R3 (Fig. 5) and the following discussions (on page 9) into the main text to further stress the importance of momentum toroidal moment.

“Finally, we calculate the reflection phase on both the front and back surfaces based on effective media theory (see parameters in SI. V). The configurations are shown in Fig. 5. We also present the reflection phase for the nodal line system as a reference. The reflection amplitudes are all unity as the selected frequency is located in the gap or right at the nodal degeneracy (Fig. 5a), whereas only the phases of reflection change with the in-plane wavevector k_x . A TM (transverse magnetic $H_z = 0$, indicated as red lines in (b)) beam is incident onto the front/back surface. Figure 5c shows the dependence of reflection phases over the in-plane wavevector for these three cases. One sees that on the front surface the reflection phase $\arg(r)$ increases by $2\pi - \Delta$ ($\Delta < 2\pi$), while on the back surface the reflection phase decreases by about Δ . The gradient of reflection phase is linked to the anomalous shift $\ell^s = -\frac{\partial}{\partial k_{\parallel}} \arg(r)$ (including Goos-Hänchen effect and Imbert-Fedorov effect)⁸. The different reflection phase distributions for the front and back surfaces will induce either negative or (tiny) positive Goos-Hänchen shift (Fig. 5d). Therefore, the Goos-Hänchen shifts with different signs on the two surfaces serve as another important signature of MTM. In addition, the striking difference of reflection phases on the front and back surfaces shows that the MTM is orientable.

Another interesting feature of a system with momentum toroidal moment is the presence of transverse spin. Fig. S10 shows that for a given frequency close to the bandgap (red lines in Fig. S10a) the corresponding electric field polarization states (Fig. S10b) are elliptically polarized. These electric fields rotate along the propagation direction in a similar way as a bicycle wheel, which is therefore called transverse spin. Although transverse spin is common for surface plasmon polaritons⁹, its realization in bulk modes remains rare¹⁰ and may bring about many interesting applications, such as transverse spin-orbital coupling, transverse spin beam shift at an abrupt interface. In addition, because the Berry curvature distribution is related to the transverse spin, those phenomena associated to transverse spin is also lined to Berry curvature, providing another way towards characterizing Berry curvature. Figure S10c gives the transverse spin defined as $Im(\vec{E}^* \times \vec{E})$, from which we clearly see the transverse spin reaches its maximum at the valley positions.”

We moved Fig. R4 to the supplementary materials.

3. PEC/PMC supported surface states

We find when setting the front and back surfaces to be PEC and PMC boundary conditions, respectively, the front and back surfaces support helical surface states. Here the PEC/PMC boundary condition serve as a mirror ($M = -1/+1$) and provides the other half Chern number⁷. This finding shows the vast difference between the momentum toroidal moment and normal nodal line.

Fig. R2. (Fig. S9 in supplementary information) Surface states (blue and red) supported by momentum toroidal moment with PEC (perfect electric conductor) and PMC (perfect magnetic conductor) boundary conditions on the front and back surfaces, respectively. a, Calculated surface states. b, Supercell with PEC/PMC boundary condition (BC) imposed on the front/back surface. c, The corresponding surface states (SS) as marked in (a).

In order to further highlight the novel topological features of momentum toroidal moment, in the revised manuscript, we have added a new section in the supplementary materials as “**Supplementary Information IV: Surface states with PEC/PMC boundary conditions**”, where Fig. R2 is involved.

Comment 2.5: • How are the field distributions in Fig. 4 obtained? Are they only theoretical or also experimental?

Response 2.5: The field distributions in Fig. 4 were simulated in the CST microwave studio. In our original near-field scanning measurement, the probe could not be inserted into the interface, therefore we couldn’t experimentally map out the 2D field distribution. Instead, we measured the interface states via line-scanning at the edges of the interface, as shown in Fig. S2c.

However, during the revision, we came up with air-waveguide configurations (Figs. R8 and 9) which enabled us to probe the field distributions on the interfaces. In the revised manuscript, we added a paragraph on page 7,

“In addition to the line-scanning method, we also construct an air-waveguide as shown in Figs. S5 and 6, where we insert the probe antenna directly into the 6-mm air-gap of Face-Face or Back-Back configuration to raster-scan the interface states. The simulation and experiment results fit well again and further confirm the presence of MTM.”

Fig. R8. (Fig. S5 in supplementary information) Face-Face air-gap interface state scanning. a, Configuration for both simulation and experiment scanning. An air-gap with thickness of 6mm is built to raster-scan the Face-Face interface states. Source and probe antennas are schematically illustrated. b/c, Numerically simulated/experimentally probed interface states along k_x . d/e, Momentum/real space interface states at different frequencies.

Fig. R9. (Fig. S6 in supplementary information) Back-Back air-gap interface state scanning. a, Configuration for both simulation and experiment scanning. An air-gap with thickness of 6mm is built to raster-scan the Back-Back interface states. Source and probe antennas are schematically illustrated. b/c, Numerically simulated/experimentally probed interface state along k_x . d/e, Momentum/real space interface states at different frequencies.

Comment 2.6: • Why exactly do the authors state that they provide a “new mechanism for the formation of Berry curvature”? It is not clear to me what “new mechanism” is supposed to mean. Isn’t this just a new form/ 3D field of the Berry curvature (although possibly interesting)?

Response 2.6: We apologize for the confusing statement.

It has been commonly accepted that in 3D systems the Berry curvatures are generated by the positive/negative Weyl points. Here “new mechanism” refers to the fact that in gapped nodal line

systems, the Berry flux is completely sourceless and it has the form of closed loops, i.e., momentum toroidal moment.

For a PT symmetric two-band nodal line, the Hamiltonian can be expressed in a general form as

$$H = f(k_x, k_y, k_z)\sigma_x + g(k_x, k_y, k_z)\sigma_z$$

We can apply the \mathbb{Z} classified topological charge to characterize it, i.e., the loop encircling the nodal line accumulates $n\pi$ ($n \in \mathbb{Z}$) Berry phase. It has been shown that one can assign each nodal line an orientation¹⁴. Once we add a small perturbation described by $h\sigma_y$, where we assume h is constant, the nodal line will be fully gapped. Then the Berry curvature distribution will follow the orientation of the original nodal line. As nodal lines can be designed in various configurations, one may manipulate Berry curvature distribution via gapping certain elaborately designed nodal lines (even more complicated configurations such as linked nodal lines, nodal knots etc).

In the revised manuscript, we have added the above argument in the discussion section (on page 10).

Comment 2.7: • In line 135, the authors mention a slight frequency shift of the experimental results as compared to theoretical considerations. This is, however, not justified. What is the reason for such a disagreement?

Response 2.7: We thank the referee for pointing out this.

Compared with the theoretical results, the frequency shift may arise from the following reasons,

1. The PCB sample has fabrication errors, while the resonance frequency is very sensitive to the size of metallic structure, such as the height of bar and length of metallic strips.
2. The dielectric constant of PCB substrate is slightly different from the labelled value.

Nonetheless, the underlying topological physics is not affected by the global frequency shift.

In the revised manuscript, we have added the discussion about the frequency shift on page 7, as copied below.

“The slight frequency shift may arise from the following reasons: The PCB sample has fabrication errors, while the resonance frequency is very sensitive to the size of metallic structure, such as the height of bar and length of metallic strips; The actual dielectric constant of the PCB substrate materials is slightly dispersive while in the simulation we consider it to be a constant.”

Comment 2.8:• There are several typos to be corrected, as well as certain formulations to be improved. I would highly suggest to find a native English speaker to read the manuscript once corrected.

e.g. 26 “exhibit” → “exhibits”

57 “physic” → “physics”

153 “re-survived” ??

159 “have possess transverse spin” ??

Response 2.8: We thank the referee for the careful reading. We have revised all of language typos and carefully proofread the whole manuscript as you suggested.

Reviewer #3 (Remarks to the Author):

Comment 3.1: The current work by B. Yang et al., discusses the formation of momentum-space toroidal moments, forming a closed-loop Berry flux. This idea is neat and interesting and can open up new directions in “topological photonics”. I have though a few major comments that the authors should consider to expand the discussions and describing better the novelty and new physics of their work.

Response 3.1: We thank the referee for considering our work to be “neat and interesting”, especially, for considering “can open up new directions in topological photonics”. We also thank the referee for the valuable comments to highly improve the presentation of our work.

Comment 3.2: 1. My main concern is that the formation of looping Berry flux does not seem to be new. As the authors describe, spin-orbit coupling happening concomitant with band inversion – in its simplest form – might lead to either Weyl points or gaped band structure. The former causes Weyl semimetals and the latter topological insulators that at their surfaces, they support distinguished electronic (or photonic) surface states (see for example the review B. Yan and C. Felser, *Annu. Rev. Condens. Matter Phys.* 2017. 8:1-19). The former as well supports “magnetic monopoles” and the latter “magnetic loops”, that one can call them a toroidal moment in the momentum space. Can only naming this feature as a toroidal moment lead to a novel discovery?

Response 3.2: We thank the referee for bringing to our notice the interesting review work on topological materials. We have cited it in the revised manuscript.

There is a key difference between our work and that mentioned in the review paper by Yan et al. In the review work the nodal line was gapped and turned into a topological insulator, which is special to spinful systems with $T^2 = -1$. In photonics, due to the lack of the Kramer’s degeneracy, one cannot realize weak/strong topological insulators in a strict sense. Therefore, our interface state configuration with the momentum toroidal moment somehow fills the gap and provides potential applications with less symmetry requirements.

In the revision turn, we found several new phenomena related to the momentum toroidal moment as clarified below.

New phenomena related to momentum toroidal moment

1. Different anomalous beam shifts on the front and back surfaces

We calculate the reflection phase on both the front and back surfaces from effective media theory. The configurations are shown in Fig. R3b. We also present the reflection phase for the

nodal line system as a reference. The reflection amplitudes are all unity as the selected frequency is located in the gap or right at the nodal degeneracy (Fig. R3a), whereas only the phase of reflection change with in-plane wavevector k_x . A TM (transverse magnetic $H_z = 0$, indicated as red lines in (b)) beam is incident onto the front/back surface. Fig. R3c shows the dependence of reflection phases over the incident angle for the three cases. One sees that on the front surface the reflection phase $\arg(r)$ increases by $2\pi - \Delta$ ($\Delta < 2\pi$), while on the back surface the reflection phase decreases about Δ . The gradient of reflection phase is linked to the anomalous shift $\ell^s = -\frac{\partial}{\partial k_{\parallel}} \arg(r)$ (Goos-Hänchen effect)⁸. The different reflection phase distributions for the front/back surface will induce either positive or negative Goos-Hänchen shift. The Goos-Hänchen shifts of different signs in the two opposite surfaces serves as another important signature of momentum toroidal moment. In addition, the striking difference of reflection phases on the front and back surfaces shows that the momentum toroidal moment is orientable. Furthermore, the difference does not depend on local surface structures (although the front and back surfaces in realistic structures have different patterns). In other words, it is one of bulk properties of the momentum toroidal moment.

Fig. R3. (Fig. 5 in the main text) Reflection phases on the front and back surfaces of the momentum toroidal moment (MTM) and nodal line (NL). a, Bulk band structure, the red line indicates the frequency we used in the calculation. b, The reflection configurations. The red arrows indicate the orientations of the MTM. Nodal line is non-orientable indicated with red line segment. The incident and reflected beams are TM (transverse magnetic, $H_z = 0$) polarized as illustrated with thin red lines. c, Reflection phases for the corresponding three different cases. d, The spatial shift for the reflected beam obtained with $-\frac{\partial}{\partial k_{\parallel}} \arg(r)$.

2. *Transverse spin in the momentum toroidal moment*

Another interesting feature of a system with momentum toroidal moment is the presence of transverse spin. Fig. R4 shows that for a given frequency close to the bandgap (red lines in Fig. R4a) the corresponding electric field polarization states (Fig. R4b) are elliptically polarized. These electric fields rotate along the propagation direction in a similar way as a bicycle wheel, which is therefore called transverse spin. Although transverse spin is common in surface plasmon polaritons⁹, its realization in bulk modes remains rare¹⁰ and may lead to many interesting applications, such as transverse spin-orbital coupling, transverse spin beam shift at an abrupt interface. In addition, because the Berry curvature distribution is related to the transverse spin, those phenomena associated to transverse spin is linked to Berry curvature, therefore providing another way for characterizing the Berry curvature. Figure R4c gives the transverse spin defined as $Im(\vec{E}^* \times \vec{E})$, from which we clearly see the transverse spin reaches its maximum at the valley positions.

Fig. R4. (Fig. S10 in supplementary information) Transverse spin in the momentum toroidal moment (MTM). a, The bulk band structure of MTM and nodal lines (NL). The red lines indicate the frequency used to plot the eigen electric fields. b, Elliptically and linearly polarized electric states in MTM and NL correspondingly. In the MTM the elliptical electric field orbits induce transverse spins pointing in the plane. c, Transverse spin defined as $\text{Im}(\vec{E}^* \times \vec{E})$, calculated with the 2nd band for both MTM and NL.

In the revised manuscript, we have added Fig. R3 (Fig. 5) and the following discussions (on page 9) into the main text to further stress the importance of momentum toroidal moment.

“Finally, we calculate the reflection phase on both the front and back surfaces based on effective media theory (see parameters in SI. V). The configurations are shown in Fig. 5. We also present the reflection phase for the nodal line system as a reference. The reflection amplitudes are all unity as the selected frequency is located in the gap or right at the nodal degeneracy (Fig. 5a), whereas only the phases of reflection change with the in-plane wavevector k_x . A TM (transverse magnetic $H_z = 0$, indicated as red lines in (b)) beam is incident onto the front/back surface. Figure 5c shows the dependence of reflection phases over the in-plane wavevector for these three cases. One sees that on the front surface the reflection phase $\arg(r)$ increases by $2\pi - \Delta$ ($\Delta < 2\pi$), while on the back surface the reflection phase decreases by about Δ . The gradient of reflection phase is linked to the anomalous shift $\ell^s = -\frac{\partial}{\partial k_{\parallel}} \arg(r)$ (including Goos-Hänchen effect and Imbert-Fedorov effect)⁸. The different reflection phase distributions for the front and back surfaces will induce either negative or (tiny) positive Goos-Hänchen shift (Fig. 5d). Therefore, the Goos-Hänchen shifts with different signs on the two surfaces serve as another important signature of MTM. In addition, the striking difference of reflection phases on the front and back surfaces shows that the MTM is orientable.

Another interesting feature of a system with momentum toroidal moment is the presence of transverse spin. Fig. S10 shows that for a given frequency close to the bandgap (red lines in Fig. S10a) the corresponding electric field polarization states (Fig. S10b) are elliptically polarized. These electric fields rotate along the propagation direction in a similar way as a bicycle wheel, which is therefore called transverse spin. Although transverse spin is common for surface plasmon polaritons⁹, its realization in bulk modes remains rare¹⁰ and may bring about many interesting applications, such as transverse spin-orbital coupling, transverse spin beam shift at an abrupt interface. In addition, because the Berry curvature distribution is related to the transverse spin, those phenomena associated to transverse spin is also lined to Berry curvature, providing another way towards characterizing Berry curvature. Figure S10c gives the transverse spin defined as $Im(\vec{E}^* \times \vec{E})$, from which we clearly see the transverse spin reaches its maximum at the valley positions.”

We moved Fig. R4 to the supplementary materials.

3. PEC/PMC supported surface states

We find when setting the front and back surfaces to be PEC and PMC boundary conditions, respectively, the front and back surfaces support helical surface states. Here the PEC/PMC boundary condition serve as a mirror ($M = -1/+1$) and provides the other half Chern number⁷. This finding shows the vast difference between the momentum toroidal moment and normal nodal line.

Fig. R2. (Fig. S9 in supplementary information) Surface states (blue and red) supported by momentum toroidal moment with PEC (perfect electric conductor) and PMC (perfect magnetic conductor) boundary conditions on the front and back surfaces, respectively. a, Calculated surface states. b, Supercell with PEC/PMC boundary condition (BC) imposed on the front/back surface. c, The corresponding surface states (SS) as marked in (a).

In order to further highlight the novel topological features of momentum toroidal moment, in the revised manuscript, we have added a new section in the supplementary materials as “**Supplementary Information IV: Surface states with PEC/PMC boundary conditions**”, where Fig. R2 is involved.

Comment 3.3: 2. In connection with the above comment, what did we learn about this toroidal moment? The fact that gaped topological insulators can support surface waves is not a new finding. Does this toroidal moment aspect lead to new physics? What kind of new features for surface waves were observed here that has yet not been reported in the literature?

Response 3.3: We thank the referee for the considerate and valuable question.

In classic waves, including photonics and phononics, the usual topological insulators protected by time-reversal symmetry with $\mathcal{T}^2 = -1$ do not exist. Although the fact that breaking nodal line may introduce topological insulator is well known in condensed matter community, it is still new

in photonics to realize helical states by gapping the nodal line. Realizing photonic topological insulator is very challenging without the Kramers degeneracy in a spinless system. Here, our structure can realize quite similar effect to the 3D helical surface waves which has not been realized in photonics due to the strict symmetry requirement.

In the revised manuscript we have added several new phenomena related to the momentum toroidal moment, please refer to **Response 3.2**.

Comment 3.4: 3. The authors discuss that the reciprocal toroidal moment has different time-space symmetry rules. What is the effect of this aspect on the surface states or bulk states? An experimental proof for this claim can indeed unambiguously support the existence of the toroidal moment.

Response 3.4: We thank the referee for the considerate and valuable question.

Different from the real space static toroidal moments, they require for breaking both T and P . In the momentum space both T and P reverse the momentum \vec{k} , which is different from the real space toroidal moment in which only P flips \vec{r} . Thus we cannot simply transfer the symmetry classification of electric/magnetic/toroidal dipole moments from real to momentum space. However, the axial- or polar- nature of various dipole moments does not depend on the space, and both the magnetic toroidal moment in the real space and the Berry curvature toroidal moment in the momentum space are polar vectors. Our theoretical and experimental analysis both support this claim.

In the revised manuscript (page 4) we have added the following paragraph.

“Different from the real space static toroidal moments, which require for breaking both T and P , here the time-reversal symmetry T (represented by $\sigma_z K$ with K being complex conjugate) is preserved while the inversion symmetry P is explicitly broken. In the momentum space both T and P reverse the momentum \vec{k} , which is different from the real space where only P flips \vec{r} . Thus we can not simply transfer the symmetry classification of electric/magnetic/toroidal dipole moments from real to momentum space. However, the axial- or polar- nature of various dipole moments does not depend on the space, and both the magnetic toroidal moment in the real space and the Berry curvature toroidal moment in the momentum space are polar vectors.”

Comment 3.5: 4. There are certain disagreements between theory and experiment. For example, it seems that in the experimental results one of the surface states is missing (Fig. 3f and i). What is the reason for this asymmetry in the band structure?

Response 3.5: We thank the referee for the question.

For the mapping of the interface states, we used the line-scanning configuration as shown in Fig. S2c (as copied below for convenience). The red antenna as source can excite the interface states will can only propagate along +y direction. When we scan the field along the line, the probe only collects the forward propagating interface states (i.e. with a positive dispersion slope), as shown in Figs. 3f and i. Interestingly, the method can directly show the positive and negative refraction

indices of interface states. For the positive refraction index, the group velocity and phase velocity both point to the $+y$ direction, and hence the probed interface state is located in the $+k_y$ region (Fig. 3i). On the other hand, the interface state with negative refraction index appears in the $-k_y$ region, meaning that the phase velocity is opposite to the energy flow (group velocity) which is still along $+y$.

Fig. R10. (Fig. S2 in supplementary information) Experimental setup for mapping bulk and Back-Back/Face-Face interface states. a, Sample fabricated by printed circuit board (PCB) in a commercial company. Face and Back sides are defined. b, Setup for probing bulk states. The source (red) is located beneath the center of the bottom layer. The component of probing electric field can be controlled by adjusting the orientation of probing antenna (purple). We used 20 unit cells when measuring bulk states. In order to show the source antenna, one quarter sample is cut. c, Interface state mapping with line-scanning the interface constructed by 10-10 unit cells. Source is indicated in red.

Comment 3.5: In general, although I have found the results beautiful and overall convincing, much more explanations about the novelty and physics of the observations here are required.

Response 3.6: We thank the referee for considering our results to be “beautiful and overall convincing”.

In the revised manuscript, we have added some discussions following the referee's kind suggestions to better describe the new physics and novelty of our work (as in our reply to the Reviewer's **Comments 3.2**).

References

- 1 Zeng, Y. et al. Electrically pumped topological laser with valley edge modes. *Nature* 578, 246-250, doi:10.1038/s41586-020-1981-x (2020).
- 2 Yang, Y. et al. Terahertz topological photonics for on-chip communication. *Nature Photonics* 14, 446-451, doi:10.1038/s41566-020-0618-9 (2020).
- 3 Wang, M. et al. Valley-locked waveguide transport in acoustic heterostructures. *Nature Communications* 11, 3000, doi:10.1038/s41467-020-16843-z (2020).
- 4 Xiao, D., Yao, W. & Niu, Q. Valley-Contrasting Physics in Graphene: Magnetic Moment and Topological Transport. *Physical Review Letters* 99, 236809, doi:10.1103/PhysRevLett.99.236809 (2007).
- 5 Xiao, D., Chang, M.-C. & Niu, Q. Berry phase effects on electronic properties. *Reviews of Modern Physics* 82, 1959-2007, doi:10.1103/RevModPhys.82.1959 (2010).
- 6 Chiu, C.-K., Chan, Y.-H. & Schnyder, A. P. Quantized Berry Phase and Surface States under Reflection Symmetry or Space-Time Inversion Symmetry. arXiv:1810.04094 [cond-mat.mes-hall] (2018).
- 7 Yao, W., Yang, S. A. & Niu, Q. Edge States in Graphene: From Gapped Flat-Band to Gapless Chiral Modes. *Physical Review Letters* 102, 096801, doi:10.1103/PhysRevLett.102.096801 (2009).
- 8 Liu, Y., Yu, Z.-M., Xiao, C. & Yang, S. A. Quantized Circulation of Anomalous Shift in Interface Reflection. *Physical Review Letters* 125, 076801, doi:10.1103/PhysRevLett.125.076801 (2020).
- 9 Bliokh, K. Y., Smirnova, D. & Nori, F. Quantum spin Hall effect of light. *Science* 348, 1448, doi:10.1126/science.aaa9519 (2015).
- 10 Peng, L. et al. Transverse photon spin of bulk electromagnetic waves in bianisotropic media. *Nature Photonics* 13, 878-882, doi:10.1038/s41566-019-0521-4 (2019).
- 11 Pendry, J. B. Negative Refraction Makes a Perfect Lens. *Physical Review Letters* 85, 3966-3969, doi:10.1103/PhysRevLett.85.3966 (2000).
- 12 Yan, B. & Felser, C. Topological Materials: Weyl Semimetals. *Annual Review of Condensed Matter Physics* 8, 337-354, doi:10.1146/annurev-conmatphys-031016-025458 (2017).
- 13 Martín-Ruiz, A. & Cortijo, A. Parity anomaly in the nonlinear response of nodal-line semimetals. *Physical Review B* 98, 155125, doi:10.1103/PhysRevB.98.155125 (2018).
- 14 Wu, Q., Soluyanov, A. A. & Bzdušek, T. Non-Abelian band topology in noninteracting metals. *Science* 365, 1273, doi:10.1126/science.aau8740 (2019).

Reviewers' Comments:

Reviewer #1:

Remarks to the Author:

The authors provided a rather detailed response that helps to better contextualize their manuscript. I particularly appreciate the link established with the 2D valley Hall effect. This addition better contextualizes the results obtained in the present work and stresses their significance.

My previous comments could be addressed to valley Hall physics as well, I agree. Also in those systems, additional crystalline symmetries are crucial to enhance the localization of the Berry curvature in momentum space and increase the robustness of the observed phenomena. Opening a gap in a system with degeneracies that carry quantized Berry curvature leads to interesting valley-Hall like phenomena in the limit of a sufficiently small energetic gap. This notion is well-established and explored particularly in 2D. This manuscript presents a natural follow-up of this line of research with the extension to nodal-lines in 3D. The studied phenomenology, though, does not present striking differences or beneficial applications compared to other valley-Hall based platforms. In particular, it shares its lack of proper topological robustness. The latter, I agree with the authors is uncommon in bosonic classical platforms and it requires careful engineering of artificial symmetries.

The toroidal configuration of Berry curvature seems a detail stemming from the original shape of the nodal line and it surely influences the details of the observed phenomenology. It does not seem to play an essential role in the emergence itself of the reported phenomenology.

There are some minor comments that I think should be addressed.

- The expression "occupied bands" should be avoided in bosonic systems lacking a well-defined Fermi surface and Pauli exclusion principle. I would use a different expression, e.g., "bands with frequencies below the bulk's band-gap".
- The supplementary material presents two different techniques used to perform the Fourier transform on the experimental data. It should always be stated which technique is used in the figures of the main text. Especially, given that the technique that neglects dissipations and provides visually more appealing results is being used.
- I still feel uncomfortable with the term "quasi-quantized". The very definition of quantization pertains to something sharp, e.g., either 0 or 1, without any other possible value in between. Something that attains values close to 1 is not quantized. I would suggest to rather specify how the Berry curvature remains localized/confined around the original profile of the nodal line for a sufficiently small gap. I consider this point rather important. Indeed, here the lack of quantization is a prominent feature of the system studied. It is different from other situations, e.g., symmetry protected topological insulators, where the lack of quantization stems from unavoidable symmetry-breaking imperfections in the experimental implementation.
- The parity anomaly quoted in the response to referees is a phenomenon that pertains to quantum systems and not classical realizations that share other phenomenology, e.g., the helical edge states.

Reviewer #2:

Remarks to the Author:

In the resubmitted manuscript, the authors have addressed all of the points that I have raised in the last report. Most importantly, the interesting aspects of the toroidal magnetic moment are more clearly explained and motivated in the manuscript. The new measurements of the field at the interfaces (comment 2.5) is also a valuable improvement, contributing to the experimental support of the theory. I still have two not resolved comments:

Response 2.3:

"Note that the valley Chern number is quasi-quantized and the quantization is exact only in the limit of diminishing bandgap. In fact, the less symmetry requirement makes valley related effects much more accessible in experimental implementation and for the universality of all kinds of waves, which is an important advantage compared with those topological phases strictly protected by symmetries."

I believe I have managed to understand the message this time, but the sentences are still very unclear. Is "limit of diminishing gap" intended to mean vanishing gap? What does the second sentence mean exactly? What does "less symmetry requirement" mean?

Comment 2.6:

I still think that "new mechanism for Berry curvature formation" is a misleading term, as Berry curvature is not necessarily linked only to Weyl points and point sources. The clarification, however, makes it more understandable.

I would support the publication of the manuscript if these two points are taken into account. As earlier, I suggest finding a native English speaker to proofread the manuscript before the final version is resubmitted.

Reviewer #3:

Remarks to the Author:

The authors have addressed my comments and the comments from all author reviewers completely. I have no other concern and would be happy to see this paper published in Nature Communications.

Response to reviewers of Nature Communications manuscript NCOMMS-20-23864A:

Reviewer #1 (Remarks to the Author):

Comment 1.1: The authors provided a rather detailed response that helps to better contextualize their manuscript. I particularly appreciate the link established with the 2D valley Hall effect. This addition better contextualizes the results obtained in the present work and stresses their significance.

My previous comments could be addressed to valley Hall physics as well, I agree. Also in those systems, additional crystalline symmetries are crucial to enhance the localization of the Berry curvature in momentum space and increase the robustness of the observed phenomena. Opening a gap in a system with degeneracies that carry quantized Berry curvature leads to interesting valley-Hall like phenomena in the limit of a sufficiently small energetic gap. This notion is well-established and explored particularly in 2D. This manuscript presents a natural follow-up of this line of research with the extension to nodal-lines in 3D. The studied phenomenology, though, does not present striking differences or beneficial applications compared to other valley-Hall based platforms. In particular, it shares its lack of proper topological robustness. The latter, I agree with the authors is uncommon in bosonic classical platforms and it requires careful engineering of artificial symmetries.

The toroidal configuration of Berry curvature seems a detail stemming from the original shape of the nodal line and it surely influences the details of the observed phenomenology. It does not seem to play an essential role in the emergence itself of the reported phenomenology.

Reply 1.1: We thank the referee for agreeing with our arguments on valley Hall physics, and for the further comments.

Although each cutting plane of the momentum toroidal moment can be regarded as a valley Hall platform, the extension to 3D is not trivial for the following reasons:

1. In our 3D system, the toroidal configuration enables all of the in-plane isotropic features, such as conical-frustum shaped domain-wall states. With the nearly in-plane isotropic electromagnetic response, the observed domain wall states provide a topological platform for realizing two-dimensional negative/positive refraction index and for exploring the associated applications.
2. Closely related to the toroidal moment is the orientability of the metamaterials, such that the front and back surfaces show different reflection phases and thus anomalous shifts. These anomalous shifts can be most conveniently explored in the 3D system.
3. In the 3D system, the Berry curvature takes the form of vector fields, whereas in 2D system, the Berry curvature is scalar. Therefore, interesting studies on the propagation of a wave packet (in both real and momentum spaces) can be carried out when introducing inhomogeneity to the 3D system.

Finally, the reviewer is correct that the configuration of toroidal Berry curvature stems from the original shape of the nodal line. This insight, first explicitly proposed in the present work, provides a powerful mean to engineer the distribution of Berry curvature in the 3D momentum space.

Comment 1.2: There are some minor comments that I think should be addressed.

- The expression "occupied bands" should be avoided in bosonic systems lacking a well-defined Fermi surface and Pauli exclusion principle. I would use a different expression, e.g., "bands with frequencies below the bulk's band-gap".

Reply 1.2: We thank the referee for the very nice suggestion. In the revised manuscript, we have updated the related phrases.

Comment 1.3: - The supplementary material presents two different techniques used to perform the Fourier transform on the experimental data. It should always be stated which technique is used in the figures of the main text. Especially, given that the technique that neglects dissipations and provides visually more appealing results is being used.

Reply 1.3: We thank the referee for pointing this out. In the revised manuscript, we have further stated it in both the main text and caption.

Comment 1.4: - I still feel uncomfortable with the term "quasi-quantized". The very definition of quantization pertains to something sharp, e.g., either 0 or 1, without any other possible value in between. Something that attains values close to 1 is not quantized. I would suggest to rather specify how the Berry curvature remains localized/confined around the original profile of the nodal line for a sufficiently small gap. I consider this point rather important. Indeed, here the lack of quantization is a prominent feature of the system studied. It is different from other situations, e.g., symmetry protected topological insulators, where the lack of quantization stems from unavoidable symmetry-breaking imperfections in the experimental implementation.

Reply 1.4: We thank the referee for pointing out this. In the revised manuscript, we have deleted the term "quasi-quantized", and stated "Berry curvature remains localized/confined around the original profile of the nodal line for a sufficiently small gap".

Comment 1.5: - The parity anomaly quoted in the response to referees is a phenomenon that pertains to quantum systems and not classical realizations that share other phenomenology, e.g., the helical edge states.

Reply 1.5: We thank the referee for pointing this out.

Reviewer #2 (Remarks to the Author):

Comment 2.1: In the resubmitted manuscript, the authors have addressed all of the points that I have raised in the last report. Most importantly, the interesting aspects of the toroidal magnetic moment are more clearly explained and motivated in the manuscript. The new measurements

of the field at the interfaces (comment 2.5) is also a valuable improvement, contributing to the experimental support of the theory. I still have two not resolved comments:

Reply 2.1: We thank the referee for his/her positive assessments of our work.

Comment 2.2: Response 2.3:

“Note that the valley Chern number is quasi-quantized and the quantization is exact only in the limit of diminishing bandgap. In fact, the less symmetry requirement makes valley related effects much more accessible in experimental implementation and for the universality of all kinds of waves, which is an important advantage compared with those topological phases strictly protected by symmetries.”

I believe I have managed to understand the message this time, but the sentences are still very unclear. Is “limit of diminishing gap” intended to mean vanishing gap? What does the second sentence mean exactly? What does “less symmetry requirement” mean?

Reply 2.2: We thank the referee for pointing out this, in the revised manuscript, we have modified the above paragraph as following,

“Note that the quantization of valley Chern number is exact only when the gap approaches to zero. In fact, the absence of symmetry requirement makes valley related topological effects more accessible in experimental implementation than those topological phases strictly protected by symmetries for all kinds of waves.”

Comment 2.3: Comment 2.6:

I still think that “new mechanism for Berry curvature formation” is a misleading term, as Berry curvature is not necessarily linked only to Weyl points and point sources. The clarification, however, makes it more understandable.

Reply 2.3: In the revised manuscript, we have deleted this term.

Comment 2.4: I would support the publication of the manuscript if these two points are taken into account. As earlier, I suggest finding a native English speaker to proofread the manuscript before the final version is resubmitted.

Reply 2.4: We thank the referee for the recommendation.

Reviewer #3 (Remarks to the Author):

Comment 3.1: The authors have addressed my comments and the comments from all author reviewers completely. I have no other concern and would be happy to see this paper published in Nature Communications.

Reply 3.1: We thank the referee for the recommendation.